# Exploiting airborne far-infrared measurements to optimise an ice cloud retrieval

Sanjeevani Panditharatne[1-3], Caroline Cox[2], Rui Song[4], Richard Siddans[2,5], Richard Bantges[1], Jonathan Murray[1,3], Stuart Fox[6], Cathryn Fox[6], and Helen Brindley[1,3]

[1]Imperial College London, London, UK
[2]RAL Space, Harwell Oxford, Chilton, UK
[3]NERC National Centre for Earth Observation, Imperial College London, London, UK
[4]NERC National Centre for Earth Observation, University of Oxford, Oxford, UK
[5]NERC National Centre for Earth Observation, RAL Space, Harwell Oxford, Chilton, UK
[6]Met Office, Exeter, UK

**Correspondence:** Sanjeevani Panditharatne (s.panditharatne21@imperial.ac.uk)

**Abstract.**

Studies have indicated that far-infrared radiances hold significant information about the microphysics of ice clouds, particularly the ice crystal habit. In support of the European Space Agency's Far-Infrared Outgoing Radiation Understanding and Monitoring mission, we perform the first retrieval on an observation of coincident upwelling far- and mid-infrared radiances taken from an aircraft above a cirrus cloud layer. Four retrievals are performed: including and neglecting the far infrared portion of the spectrum, and assuming two different habit mixes. Results are compared to in-situ measurements of the cloud optical thickness, cloud top height, cloud effective radius, and habit distributions. We find that despite the known limitations of ice cloud optical property models, all the retrievals show agreement within the in-situ measurements of the cloud optical thickness, cloud top height, and cloud effective radius. However, the inclusion of the far-infrared enables a distinction between two different habits that is not possible using only mid-infrared channels. Furthermore, in this case study, the uncertainty in the retrieval of cloud top height and cloud optical thickness halves with the inclusion of the far infrared. As with other studies, we also see an additional degree of freedom for the temperature and water vapour retrievals. Our study highlights the need for the improvement of current ice cloud optical models, with the radiance residuals from the converged retrievals still exceeding the instrument uncertainty within the far-infrared. However, it provides observational support for the theoretical improvement that far-infrared observations could bring to retrievals of ice cloud properties.

## 1 Introduction

Cirrus clouds cover ∼30 % of the Earth, and there is strong evidence suggesting that they have a net warming effect on the atmosphere through enhanced trapping of the Earth's Outgoing Longwave Radiation (OLR) (Stubenrauch et al., 2006; Sassen et al., 2008). This is particularly true for high, optically thin cirrus (Hong et al., 2016). Studies suggest that anthropogenic activities are influencing the micro- and macrophysics of cirrus clouds, with surface warming increasing the number of these high and optically thin clouds (Haywood et al., 2009; Zhou et al., 2014; Zhao et al., 2019; Singh et al., 2024).

Currently, there are large disparities between climate model predictions of the net radiative forcing and feedback processes of cirrus clouds (Forster et al., 2021). This is in part due to the variation in ice crystal microphysics and a limited understanding about their formation and development (Fu et al., 2017). In comparison to water clouds, the habit of each crystal introduces additional complexity for the calculation of scattering and absorption of incident radiation (Baran et al., 2014). Poorly constraining these habits can also result in significant errors in our understanding of the other cloud microphysical properties (Mishchenko et al., 1996; Yang et al., 2015).

Active sensors have been used to estimate ice crystal habits from observations, however, these observations have a low spatial coverage, and are limited by the geometric thickness of the cloud (Saito et al., 2017; Pfreundschuh et al., 2022). Using passive remote sensors that cover the mid- and near-infrared can offer greater spatial coverage, but the contributions of surface reflectance, particularly over land, create challenges for retrieving properties from optically thin cirrus (Baran et al., 1998; Chepfer et al., 2002).

Theoretical studies have indicated that the far-infrared region (100-667 $cm^{-1}$) of the Earth's OLR is highly sensitive to cirrus microphysics, particularly the ice crystal habit (Maestri and Rizzi, 2003; Baran, 2005, 2007). Due to technical limitations, the first systematic observations of spectrally resolved far-infrared (FIR) radiances at the top of the atmosphere (TOA) have only begun recently, with the recent launch of NASA's Polar Radiant Energy in the Far-InfraRed Experiment in summer 2024 (L'Ecuyer et al., 2021). Coverage of the far-infrared with higher spectral resolution and radiometric accuracy will follow with the launch of ESA's Far-infrared Outgoing Radiation Understanding and Monitoring (FORUM) mission in late 2027. The FORUM mission aims to measure the Earth's spectrally resolved OLR using the FORUM Sounding Instrument (FSI) over the spectral range of 100-1600 $cm^{-1}$ with a spectral resolution greater than 0.5 $cm^{-1}$ and a target radiometric accuracy of 0.1 K at $3\sigma$ (Palchetti et al., 2020).

Simulation studies have indicated that including a few far-infrared channels in current retrievals from existing mid-infrared (MIR) spaceborne radiometers could significantly reduce the uncertainties of cloud optical thickness (COT), cloud effective radius (CER), and cloud top height (CTH), particularly in polar regions and the upper tropical troposphere (Libois and Blanchet, 2017). Some retrievals have also been performed on observations of downwelling far-infrared radiances in the presence of ice clouds. However, these were limited by uncertainties in the water vapour and temperature profiles that would be less dominant when looking at upwelling radiances (Di Natale et al., 2017; Maestri et al., 2014)

At present, there are no published retrievals from observed upwelling far-infrared radiances in the presence of cirrus cloud, and only two published studies have attempted to replicate upwelling far-infrared radiances measured under these conditions. Cox et al. (2010) was limited by large uncertainties in the atmospheric state and a lack of instrumentation capable of measuring small ice particles. Bantges et al. (2020) found that current ice cloud bulk optical property models were unable to simultaneously simulate radiances across the mid- and far-infrared within the instrument uncertainty. Despite these findings, these bulk optical property models are still commonly used to characterise ice clouds from far-infrared radiances (Magurno et al., 2020; Di Natale and Palchetti, 2022; Peterson et al., 2022).

This work uses the observation described in Bantges et al. (2020) to perform the first retrieval of cirrus properties from airborne measurements covering the far- and mid-infrared. In preparation for the FORUM mission, we adapt this observation

to mimic the expected FSI spectral characteristics before performing retrievals using the RAL Infrared Microwave Sounding Scheme (IMS). We aim to investigate two key areas of interest that have arisen from prior work. Given the sensitivity of the far-infrared to ice crystal habit, we explore if including far-infrared channels in a retrieval allows us to distinguish between two different models of ice crystal habits developed by Yang et al. (2013) and Baum et al. (2014): the Solid Columns (SC) and General Habit Mix (GHM). While the SC model only includes solid columns, the GHM model incorporates plates, droxtals, hollow and solid columns, hollow and solid bullet rosettes, an aggregate of solid columns, and a small and large aggregate of plates. These two models were selected based on their spectral distinction in the far-infrared (Bantges et al., 2020). There are currently known limitations in these models in their representation within the far-infrared, and so our second aim is to assess the impact of these on the retrieved values and radiance residuals.

To do this, we first perform retrievals of visible COT, CER, and CTH using a fixed temperature and gaseous profile on the observation and on a comparable simulation to evaluate the significance of cloud approximations used within the retrieval scheme. We then perform simultaneous retrievals of temperature, water vapour, visible COT, CER, and CTH and evaluate the results against in-situ measurements of the cloud and atmospheric state.

The layout of the paper is as follows: in Section 2 we outline the instrumentation used to measure both the upwelling radiance and state properties. Section 3 details the retrieval scheme and methodology used. Section 4 contains the retrievals of COT, CER, and CTH where the temperature and water vapour profile are fixed. Section 5 contains the simultaneous retrieval of temperature, water vapour, and cloud properties. Conclusions are drawn in Section 6.

## 2 Observational Data

### 2.1 The B895 Flight

The B895 flight took place on 13 March 2015 as part of the Cirrus Coupled Cloud-Radiation Experiment (CIRCCREX) campaign which had the overarching goal of improving our understanding of cirrus cloud microphysics (Pickering et al., 2015). The main objective of the flight was to simultaneously measure upwelling radiances in both the mid- and far-infrared using the Tropospheric Airborne Fourier Transform Spectrometer (TAFTS) (Canas et al., 1997) and the Airborne Research Interferometer Evaluation System (ARIES) (Wilson et al., 1999) above a well characterised cirrus cloud. The flight was performed using the Facility for Airborne Atmospheric Measurements (FAAM) British Aerospace (BAe) 146 aircraft, and was based out of Prestwick, Scotland.

Figure 1 shows the flight track overlaid on top of the L2 COT from the MODIS (Moderate Resolution Imaging Spectroradiometer) Terra overpass at 10:40 UTC. The FAAM aircraft performed three straight and level runs (SLR) at an altitude of 9.4 km overflying a decaying band of cirrus cloud associated with an occluded front. We focus on a radiance observation taken at 09:48:39 UTC during the first SLR (09:33 to 09:58 UTC), where the plane was flying in a north-westerly direction. At this time, the atmosphere above the aircraft was cloud-free, limiting the amount of downwelling radiation available for back-scatter into the radiometric instrumentation from the underlying cirrus.

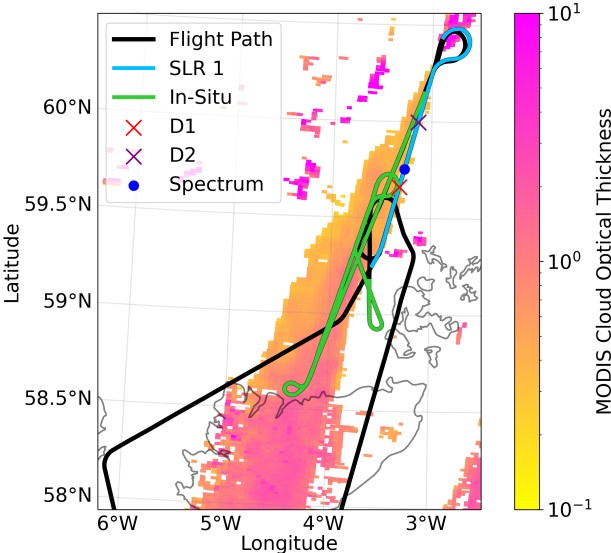

**Figure 1.** The flight track (black) for the B895 flight that took place on 13 March 2015 plotted on the cloud optical thickness product from a MODIS Terra satellite overpass at 10:40 UTC. The first straight and level run (SLR 1) is shown in light blue, and the location of the selected radiance observation is marked with a blue circle (09:48:39 UTC). Dropsondes released during SLR 1 are marked with a cross. Approximately 50 minutes after the radiance observations were taken, the aircraft descended through the cloud taking in-situ measurements following the light green path between 10:32 and 11:32 UTC.

The aircraft was also equipped with the Airborne Vertical Atmospheric Profiling System, which released two Vaisala RD94 dropsondes at 09:47:48 and 09:50:42 UTC during the first SLR. Given that these dropsondes use the same humidity sensor as the Vaisala RS92 radiosonde, we estimate the sensor calibration uncertainty was $\pm5\%$ of the measured relative humidity with an absolute offset of $\pm0.5\%$. The production variability uncertainty was $\pm1.5\%$ above 10% or $\pm3\%$ for relative humidity below 10% (Miloshevich et al., 2009). However, Fox et al. (2019) found discrepancies between simulated and observed microwave brightness temperatures in clear-sky conditions for other flights around the same time as B895 when using the measured dropsonde profiles, and this was attributed to a dry bias in the dropsondes. No equivalent information is available for the temperature sensor, but there is a manufacturer-quoted repeatability of 0.2 K

After completing the other two SLRs, the plane descended through the cloud and measured the microphysics using on-board instrumentation between 10:32 and 11:32 UTC. During these manouevres the cirrus remained relatively stationary, with satellite imagery indicating that it had dispersed by 13:00 UTC (O'Shea et al., 2016).

## 2.2 Cloud Instrumentation

There were a number of instruments onboard the FAAM aircraft used to characterise the cirrus cloud that will be used to assess the retrievals performed here. These include an elastic backscatter lidar (used to derive a reference for the COT and CTH), a

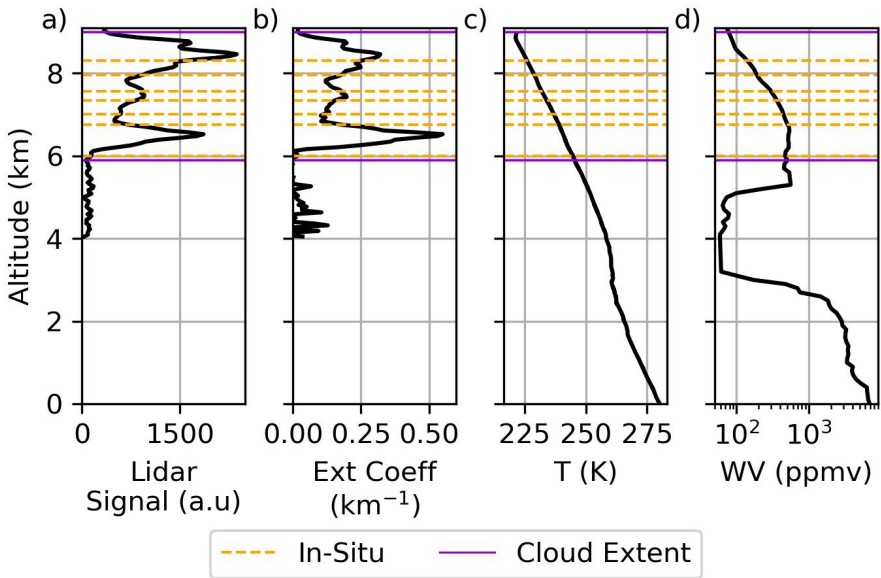

**Figure 2.** (a) The range-corrected lidar signal of the cloud measured at 09:48:35 UTC. (b) The lidar-derived extinction coefficient of the cloud measured at 09:48:35 UTC. The (c) temperature and (d) water vapour profiles measured by dropsonde 1 at 09:47:48 UTC. The edges of the cloud extent are marked in purple, and the altitudes at which in-situ measurements of cloud habit and PSDs have been characterised are shown with dashed orange lines (6, 6.8, 7, 7.3, 7.6, 8.0, and 8.3 km).

2-DS probe (used to derive a reference for the CER), a 3-View Cloud Particle Imager (3V-CPI) (used to derive the ice crystal habits in O'Shea et al. (2016)), and a CIP 100 and a holographic cloud probe (HALOHolo) (used to measure the particle sizes). Further details about each of the instruments and their measurements are provided below.

A Leosphere ALS450 355 nm elastic backscatter lidar was installed on the aircraft (Marenco, 2010). This provided information on the cloud vertical extent, ice volume extinction profiles at 355 nm from the range-corrected backscatter profiles following the two-stage process of Marenco et al. (2011), and a vertical profile of the particle extinction coefficient using the method described in Fox et al. (2019). Raw lidar profiles were recorded with a vertical resolution of 1.5 m and an integration time of 2 s. The range-corrected backscatter profile measured by the lidar at 09:40:35 UTC (4 seconds before the radiance observation) is shown in Figure 2a alongside the lidar-derived extinction profile (Figure 2b), and the temperature and water vapour profiles measured by dropsonde 1 (Figure 2c and d). This lidar-derived extinction coefficient profile has been previously used to derive the COT, however, due to poor instrument alignment, there were issues with the depolarisation calibration which limited the quality of the estimated COT (Fox et al., 2019).

To mitigate these issues, we employ a simplified retrieval approach to derive the COT rather than the full vertical extinction profile. This approach considers the power of the attenuated Rayleigh backscatter range-corrected signals below 6 km in cirrus-free ($P_{ref}$) and cirrus ($P_{att}$) conditions. It is assumed that the molecular extinction is constant along the short flight path, and so

the signal variation can be attributed to the ice crystals in the cirrus cloud. The presence of cirrus reduces the attenuated $P_{att}$ relative to $P_{ref}$ through two-way transmission and simplifies the COT retrievals to:

$$\text{COT}_{355\,\text{nm}} = -\frac{1}{2}\ln\left(\frac{P_{att}}{P_{ref}}\right) \tag{1}$$

where the factor of 2 accounts for the two-way transmission through the cirrus layer. In this work, $P_{ref}$ is calculated from the average of one minute of backscatter signals between 4 and 6 km at 09:33 UTC where the atmosphere was cloud-free below the aircraft. $P_{att}$ represents the backscatter signals from the remainder of the SLR that have then been similarly vertically averaged between 4 and 6 km for each of the 2 s measurement intervals. This means we now state the average lidar-derived COT for the one-minute period surrounding the radiance observation as 0.8±0.1, with the uncertainty representing the standard error in the mean COT within this time period.

The particle size distributions (PSDs) and ice crystal habits were measured using a series of probes that included a 2-DS, a 3V-CPI, a CIP 100 and a HALOHolo (Lawson et al., 2006b; O'Shea et al., 2016). The 2-DS and inlet edge of the 3V-CPI probes were fitted with antishatter tips to reduce a bias towards smaller crystals. O'Shea et al. (2016) presented an analysis of the in-situ measurements taken during the B895 flight, with both PSD and habits divided into the different temperature regions shown in Figure 2. Each set represents the average of a run that took between approximately 5 and 10 minutes through sections of the cloud. The ice crystals were found to range from approximately 10 µm (lower limit of the 2-DS probe) to 700 µm. The best-fit distribution for the PSDs was generally found to be a bimodal Gaussian and will be used to estimate the CER in Section 2.2.1.

The ice crystal habits within the cloud were determined from CPI images of crystals larger than 50 µm using an automatic habit recognition algorithm outlined in O'Shea et al. (2016). Particles were found to be a mix of predominantly aggregates ($\sim 30$ %) and droxtals ($\sim 25$ %), with some rosettes ($\sim 15$ %) and columns ($\sim 10$ %) throughout the cloud. This is more similar to the constituents of the GHM model than the SC model, and so we expect the Yang et al. (2013) and Baum et al. (2014) GHM model to be a better representation of the ice crystal habits within the cirrus cloud.

## 2.2.1 Estimating the CER

The in-situ measurements of the cloud PSDs and ice crystal habits analysed in O'Shea et al. (2016) can be used to provide an approximation of the CER. The CER used in the Baum et al. (2014) and Yang et al. (2015) bulk optical property models is defined by the following relation:

$$\text{CER} = \frac{3}{2}\frac{V_{tot}}{A_{tot}} \tag{2}$$

where $V_{tot}$ and $A_{tot}$ are the total volume and total projected area, respectively, of the ice crystals within the bulk cloud (Baum et al., 2014). These are calculated for each type of ice crystal habit ($h$) using equations 3 and 4:

**Table 1.** The cloud effective radius (CER) calculated from the PSDs measured by the Cloud 2-DS probe, and the habit distributions measured by the CPI for temperature regions shown in Figure 2. The temperatures have been converted to altitudes using the dropsonde measurements (Figure 2c.), and have been used to calculate the approximate times and distances since and from the radiance observation.

| Temperature (K) | Altitude (km) | Time Since Rad Obsv (min) | Dist to Rad Obsv (km) | CER ($\mu$m) |
|---|---|---|---|---|
| 229 | 8.0 | 55 | 72 | 56±1 |
| 234 | 7.3 | 65 | 126 | 20±1 |
| 237 | 7.0 | 75 | 98 | 29±1 |
| 239 | 6.8 | 80 | 38 | 47±6 |
| 245 | 6.0 | 95 | 71 | 42±5 |

$$A_{\text{tot}} = \sum_{h=1}^{M} \left[ \int_{D_{\text{min}}}^{D_{\text{max}}} A_h(D) f_h(D) n(D) dD \right] \tag{3}$$

and

$$V_{\text{tot}} = \sum_{h=1}^{M} \left[ \int_{D_{\text{min}}}^{D_{\text{max}}} V_h(D) f_h(D) n(D) dD \right] \tag{4}$$

where $M$ is the number of ice crystal habits observed in the cloud, $D_{\text{min}}$ and $D_{\text{max}}$ are the minimum and maximum sizes in the PSD, $A_h(D)$ and $V_h(D)$ are the projected area and volume of a specific crystal with habit and size ($D$), $f_h(D)$ is the fraction of ice crystals that have a given habit at each size, and $n(D)$ is the size number distribution.

Following the findings of O'Shea et al. (2016), we find $n(D)$ by fitting the PSDs measured by the Cloud 2-DS probe with bimodal Gaussian distributions using the non-linear least squares method (Lawson et al., 2006a; Zhao et al., 2011; Jackson

et al., 2015). The fitted PSDs and parameters are shown in Figure S1 and Table S1 in the Supplement. Due to the dominance of smaller crystals (<30 μm), as observed in O'Shea et al. (2016), we are unable to fit the PSDs measured at 232 and 226 K (7.5 and 8.3 km) with a bimodal Gaussian distribution, and so they have been excluded from this work. Similarly, the habit types and fractions are those that were derived from the CPI measurements using the habit recognition algorithm presented in (O'Shea et al., 2016), while $A_h(D)$ and $V_h(D)$ are derived from the Yang et al. (2013) database for each observed habit. The uncertainty

presented here for the CER is calculated using the relative standard deviation of the fitting parameters, and so represents one standard deviation in the fitting of the bimodal Gaussian distribution to the PSD. We note that larger uncertainties are expected in the PSDs themselves, with some work suggesting that the smaller mode of the distribution is linked to diffraction effects from the instrumentation (O'Shea et al., 2021).

Table 1 shows the estimated CER values for each altitude with approximate times and distances relative to that of the radiance observation. These range from 20-56 μm with an uncertainty of up to 12 % that increases closer to the base of the cloud, and were taken up to 95 minutes after the time of the radiance observation. The mean CER weighted by the individual uncertainties is 32 μm with a weighted standard deviation of 14 μm that represents the spread of CER measured within the cloud.

## 2.3 Radiometric Observations

### 2.3.1 Radiometric Instrumentation

The Tropospheric Airborne Fourier Spectrometer (TAFTS) is a four-port Martin-Puplett interferometer (Canas et al., 1997). Measurements are made at the output ports by pairs of detectors, each containing a "longwave" and "shortwave" detector made of GeGa and SiSb respectively. The instrument has two pairs of blackbody calibration targets held at ambient temperature and 323 K. There is an internal calibration before scans, with a single nadir scan taking $\sim$1.5 s and having an angular field of view of 1.6°. TAFTS has a spectral range of nominally 80-300 $cm^{-1}$ (longwave channel) and 330-600 $cm^{-1}$ (shortwave channel) with a sampling of 0.06 $cm^{-1}$ and a nominal spectral resolution of 0.12 $cm^{-1}$. The uncertainty on a single TAFTS spectrum is composed of the random noise (Figure 3a) and calibration uncertainty (Figure 3b).

The Airborne Research Interferometer Evaluation System (ARIES) uses a Michelson-type configuration with a HgCdTe photodetector for the "longwave" channel and a InSb photodetector for the "shortwave" channel (Wilson et al., 1999). It also has two temperature-controlled blackbody targets and performs periodic calibrations during its measurement sequence. A single ARIES scan takes $\sim$0.25 s and has an angular field of view of 2.5°, with an optical path difference of 1.01 cm. The instrument spectral range covers 550-1800 $cm^{-1}$ (longwave channel) and 1700-3000 $cm^{-1}$ (shortwave channel), with a spectral resolution of 1 $cm^{-1}$ and a spectral sampling of 0.42 $cm^{-1}$. Only observations from the longwave channel are considered here as the shortwave channel exceeds the FSI spectral range. The uncertainty on a single ARIES spectrum is composed of the random noise (Figure 3a) and calibration error (Figure 3b). Due to the faster scan time of ARIES, the 6 ARIES scans neighbouring the selected TAFTS scan at 09:48:39 UTC are averaged for this work, and the appropriately reduced random noise is also shown in Figure 3a.

### 2.3.2 Creating FORUM-aircraft Spectra

As this study is in support of the FORUM mission, we adapt the observed radiances to mimic the FSI instrument line shape (ILS) to create 'FORUM-aircraft' observations. The FSI instrument line shape is modelled using the strong Norton-Beer apodisation resulting in a spectral resolution of 0.6 $cm^{-1}$ with a sampling of 0.3 $cm^{-1}$. We note that the ground footprint for the FSI (15 km) is greater than for TAFTS ($\sim$0.26 km) and ARIES ($\sim$0.41 km). The apodised noise-equivalent-differential-temperature (NEDT) and target absolute radiometric accuracy (ARA) for the FSI are also shown in Figure 3, and suggest that the combined FSI uncertainty is up to 1 K smaller than the instrumentation used in this work between 200 and 1300 $cm^{-1}$. While this will affect the significance of including the FIR, Panditharatne et al. (2025) found that for clear sky retrievals of

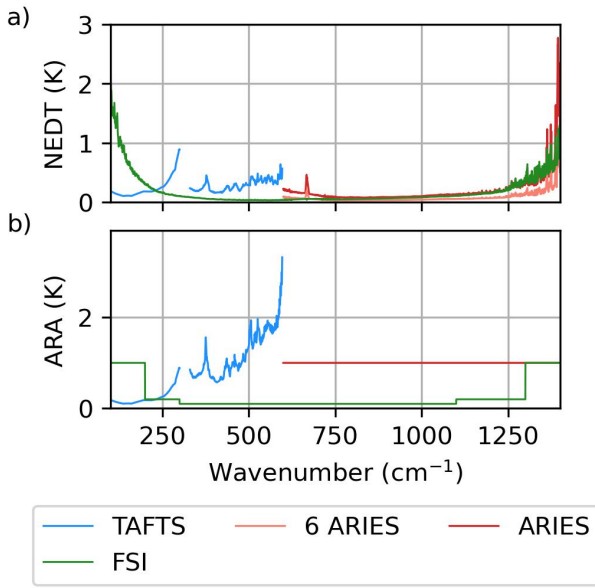

**Figure 3.** (a) The noise-equivalent-differential-temperature (NEDT) and (b) absolute radiometric accuracy (ARA) for TAFTS, ARIES, 6 ARIES scans, and target values for the FSI based on the selected radiance observation.

temperature and water vapour, the simulated FSI and FORUM-aircraft spectra produced similar results, suggesting a retrieval from a FORUM-aircraft observation is indicative of its FSI counterpart given a homogeneous scene.

To create the FORUM-aircraft observations, we follow the process in Panditharatne et al. (2025). The TAFTS ILS is approximated using an apodisation function, and so this is first deconvolved from the observed spectrum. The FSI instrument line shape is then applied to both the TAFTS and ARIES observations. This introduces an uncertainty into the observed FORUM-aircraft spectrum as a result of the original instrument's spectral characteristics. To quantify this uncertainty, we evaluate the effect on a high-resolution simulated spectrum that has a similar surface and atmospheric profile to our selected observation.

This high-resolution spectrum is simulated using the combined LBLDISv3.0 model that is composed of the Line-by-Line Radiative Transfer Model (LBLRTMv12.11) and DIScrete Ordinate Radiative Transfer (DISORT) code (Clough et al., 2005; Stamnes et al., 2000; Turner and Holz, 2005). LBLRTM is used to calculate the wavelength-dependent optical depths for every layer between the surface and the aircraft and uses the MT_CKDv3.5 continuum model (Mlawer et al., 2019). These layer optical depths are then input into DISORT to numerically compute the multiple scattering caused by the ice crystals using 16 streams.

We use the temperature and water vapour profile derived from dropsonde 1, and the $O_3$ profile from co-located ERA-5 data. Profiles for $CO_2$, $CH_4$, and other trace gases are set to the US 1976 mid-latitude winter profile that has been scaled to present-day concentrations using data from Mace Head (Dlugokencky et al., 2019). The surface skin temperature is set to 280.2 K, which was derived in Bantges et al. (2020) based on the dropsonde profile and collocated ERA-I data. The spectral surface

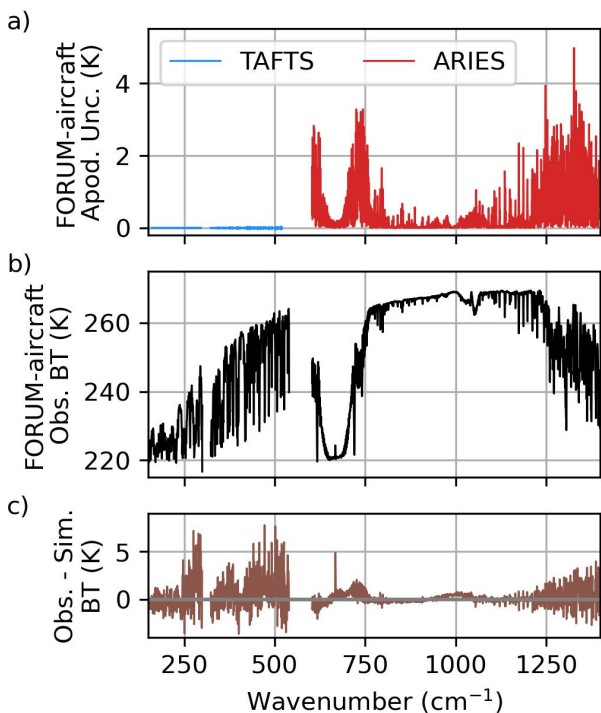

**Figure 4.** (a) The brightness temperature (BT) residual between the LBLDIS simulation of the TAFTS and ARIES observations during the B895 flight that has had the FORUM apodisation directly applied and been made to look like TAFTS or ARIES first (FORUM-aircraft simulation) as outlined in Section 2.3.2. The TAFTS residuals remain less than 0.05 K and can be seen in the Supplement for clarity. (b) The FORUM-aircraft observation. (c) The BT residuals between the FORUM-aircraft observation and simulation.

emissivity was calculated using the Masuda et al. (1988) model above 769 $\text{cm}^{-1}$ as the radiance observation was taken over the ocean. Below 769 $\text{cm}^{-1}$, the surface emissivity was found to have a negligible effect on the spectrum given the strong water vapour absorption below the cloud in the far-infrared (Bantges et al., 2020). Therefore, we fix it to 0.99. Based on the
results of the minimisation process in Bantges et al. (2020), we use the lidar-derived extinction coefficient profile (Figure 2b) scaled to output a COT at 355 nm of 0.82, a CER of 34 µm and assume the GHM model.

   This high-resolution spectrum is treated in two ways. First, it has the FSI apodisation directly applied to it, creating an FSI simulated spectrum. Second, the high-resolution spectrum is modified to look like TAFTS and ARIES before the FSI apodisation is applied to emulate the process applied to the observation. This creates a FORUM-aircraft simulated spectrum
which will be used for later analysis in Section 4.1. The residual between the FSI simulated spectrum and the FORUM-aircraft simulated spectrum is shown in Figure 4a. As in Panditharatne et al. (2025), due to the coarse sampling of the ARIES measurement, this uncertainty exceeds the instrument uncertainty in the $CO_2$ 15 micron band where it can reach up to 3.3 K, as well as above 1200 $\text{cm}^{-1}$ where it exceeds 4 K. The combination of the NEDT, ARA (Figure 3), and this apodisation uncertainty will be used to constrain the final retrieval.

The final FORUM-aircraft observation is shown in Figure 4b, with the difference to the FORUM-aircraft simulation in Figure 4c. The simulation is generally within 5 K of the observation, excluding selected channels near the TAFTS band edges, and will be used to explore the expected behaviour of the retrieval.

## 3 The Retrieval Method

### 3.1 The RAL Infrared Microwave Sounding Retrieval Scheme

The RAL Infrared Microwave Sounding (IMS) retrieval scheme simultaneously retrieves vertical profiles of atmospheric temperature and gases, along with surface skin temperature, surface spectral emissivity and cloud parameters (Rodgers, 2000; Siddans, 2019). Recent work has extended it for use on the FSI for clear sky retrievals (Panditharatne et al., 2025), and it has also been used for retrievals of aerosols and in the presence of clouds (Kloss et al., 2022; Trent et al., 2023).

IMS uses the optimal estimation method from Rodgers (2000) to fit an observed spectrum (the measurement vector, $\mathbf{y}$) by iteratively perturbing the retrieval targets (the state vector, $\mathbf{x}$). Estimations of $\mathbf{y}$ are calculated from adjusted values of $\mathbf{x}$ using a forward model, $\mathbf{F(x)}$, which in this case is a radiative transfer model. Prior knowledge of the state is contained in the a-priori state vector, $\mathbf{x}_a$, with covariance, $\mathbf{S}_a$, representing the vertical variability and correlation of the profile. These are both used to constrain the retrieval. Similarly, the measurement covariance, $\mathbf{S_y}$, represents the uncertainty in the measurement.

Iterations are based on the Levenburg-Marquardt (LM) method (Marquardt, 1963):

$$\mathbf{x}_{i+1} = \mathbf{x}_i + \left( \mathbf{K}_i^T \mathbf{S_y}^{-1} \mathbf{K} + \mathbf{S}_a^{-1} + \gamma_i \right)^{-1}$$
$$\left[ \mathbf{K}_i^T \mathbf{S_y}^{-1} \left( \mathbf{y} - \mathbf{F}(\mathbf{x}_i) \right) - \mathbf{S}_a^{-1} \left( \mathbf{x}_i - \mathbf{x}_a \right) \right] \tag{5}$$

where $i$ is the iteration, $\gamma$ is the LM parameter controlling the magnitude of the state vector perturbation and is initially set to 0.001, and $\mathbf{K}_i$ is a Jacobian matrix of partial derivatives of the forward model output to elements of the state vector.

The fit optimisation is based on minimising the cost, $\chi^2$, with the first and second terms on the right-hand side of equation 6 corresponding to the measurement and state cost, respectively:

$$\chi^2 = (\mathbf{y} - \mathbf{F}(\mathbf{x}))^T \mathbf{S_y}^{-1} (\mathbf{y} - \mathbf{F}(\mathbf{x})) + (\mathbf{x} - \mathbf{x}_a)^T \mathbf{S}_a^{-1} (\mathbf{x} - \mathbf{x}_a) \tag{6}$$

The Radiative Transfer for TOVS v12 (RTTOVv12) fast radiative transfer model is used as the forward model in the IMS scheme. It takes inputs of temperature, water vapour, ozone, carbon dioxide, cloud fraction, cloud effective diameter, and cloud ice water content (IWC) on 101 fixed pressure levels from the surface to the top of the atmosphere.

RTTOVv12 uses the strong Norton-Beer apodisation to simulate FSI spectra covering 5000 channels across the FSI spectral range. Cloudy radiances are calculated using the Chou approximation, where the optical depths associated with atmospheric emission and absorption (referred to here as clear sky optical depths) are scaled to account for the cloud particle interac-

tions (Chou et al., 1999). Both of these are parameterised into coefficients to increase operational speed (Haiden et al., 2018; Saunders et al., 2017).

To calculate clear sky optical depths, we use specialised RTTOVv12 coefficients that have been developed using LBLRTMv12.11, to simulate upwelling FORUM-aircraft spectra at common flying altitudes (Panditharatne et al., 2025). For the ice cloud interactions, the absorption coefficient, scattering coefficient, and backscattering parameter used for the Chou approximation have been calculated from the Yang et al. (2013) and Baum et al. (2014) GHM and SC bulk optical property models for ice water content (IWC) from 0.5E-5 to 0.1 $gm^{-3}$ and CER from 5-60 μm. The use of the Chou approximation to simulate cloudy radiances significantly reduces the operational speed of the retrieval, however it leads to a positive bias within the far-infrared that increases for smaller crystals (Martinazzo et al., 2021).

Within IMS, the ice cloud is modelled as a Gaussian with a standard deviation of 1 km. The visible COT is calculated from the IWC and CER within IMS using the following approximation from Wang et al. (2011):

$$\text{COT}_{\text{vis}} = \int\limits_{\text{CBH}}^{\text{CTH}} \frac{3\text{IWC(z)} < Q_{e,\text{vis}} >}{4\rho_{\text{ice}}\text{CER}} dz \tag{7}$$

where $< Q_{e,\text{vis}} >$ is the bulk extinction efficiency at visible wavelengths, $\rho_{\text{ice}}$ is the density of ice, and the integral is performed from the Cloud Base Height (CBH) to the CTH. We assume $< Q_{e,\text{vis}} >= 2$ as we can assume the average particle is much larger than the wavelength at visible wavelengths.

### 3.2 Retrieval Targets and Constraints

In this work, we perform simultaneous retrievals of COT, CER, CTH, temperature and water vapour. The a-priori and covariance for the cloud retrieval targets are 9±3 km for CTH and 40±10 μm for CER. COT is stored in log form to prevent it from reducing below zero, and it is assumed there is minimal cloud at first, resulting in an a-priori of $\ln(0.01) \pm 10$.

We use hourly co-located ERA-5 reanalysis data as the a-priori for temperature and water vapour, and the covariance is a 2-dimensional matrix derived from the differences between the zonal mean of ERA-5 profiles for three days (17 April, 17 July, and 17 October 2013) (Siddans, 2019). In all the retrievals performed in this work, the initial guess is equivalent to the a-priori, and the vertical $O_3$, $CO_2$, $CH_4$ and $N_2$ profiles, as well as the surface skin temperature and surface emissivity, are fixed to the values outlined in Section 2.3.2.

The measurement covariance (Figure 5) is assumed to be fully correlated and is composed of the NEDT and ARA for a single TAFTS and 6 ARIES scans (Figure 3), combined with the apodisation uncertainty described in Section 2.3.2 (Figure 4a).

The retrieval is performed using selected channels rather than for the entire spectral range to minimise operational time and avoid selecting channels with high measurement uncertainty. Figure 6 shows the 200 channels selected in Panditharatne et al. (2025) to optimise retrievals of temperature and water vapour from FORUM-aircraft observations.

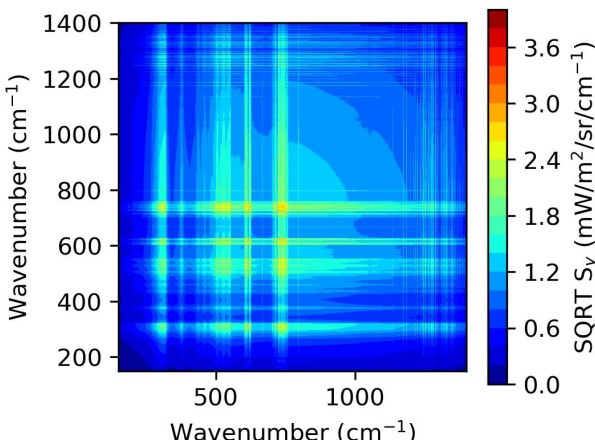

**Figure 5.** The square-root of $\mathbf{S_y}$ used in the retrievals in radiance units. This is a combination of the instrument NEDT, ARA, and uncertainty introduced through the FORUM-aircraft apodisation process shown in Figure 3 and 4a.

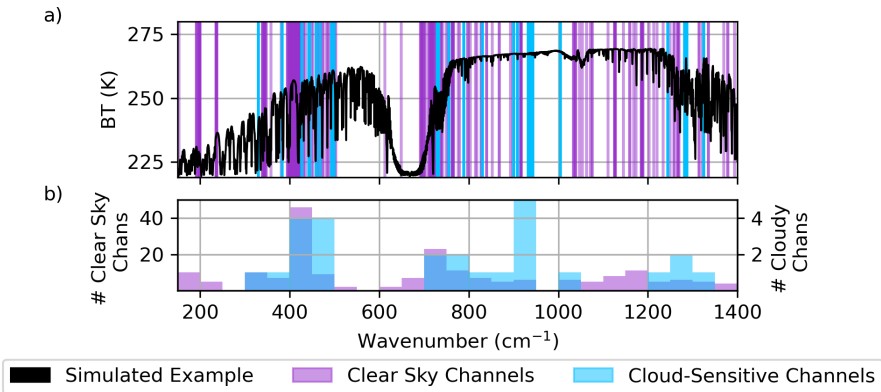

**Figure 6.** (a) The channels selected for retrievals using the FORUM-aircraft configuration of IMS on top of an example spectrum in brightness temperature (BT), with (b) the number of channels in each 50 $\mathrm{cm}^{-1}$ bins. The 200 channels selected in Panditharatne et al. (2025) for optimised retrievals of temperature and water vapour are shown in purple. The additional 26 channels that have been selected to optimise retrievals of ice cloud properties are shown in blue.

The absorption of water vapour dominates the FIR, and as a result, there is a significant overlap in this region between spectral sensitivity to tropospheric water vapour and ice cloud properties. To enable a clearer distinction between habits, 26 additional channels have been included that were not in the initial 200 but are sensitive to ice cloud properties for this specific observation (Figure 6). We anticipate that they are also sensitive to water vapour and temperature.

These additional 'cloud-sensitive' channels were selected using a database of FORUM-aircraft spectra simulated using RTTOVv12 that vary in COT (0.1-2), CER (5-60 μm), and ice crystal habit (GHM and SC models) but all have the same

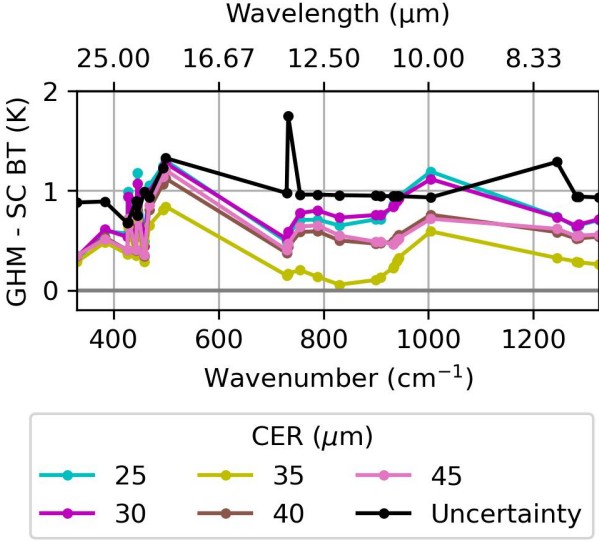

**Figure 7.** The BT residuals between the RTTOVv12 FSI simulated spectra assuming the General Habit Mix (GHM) and Solid Columns (SC) databases for example CER values and an ice water path of $\sim$28 gm$^{-2}$ derived from the lidar measurements. The remainder of the atmospheric and surface profile are as described in Section 2.3.2. The black line represents the instrument and apodisation uncertainty and is the square-root-diagonal of the covariance matrix in Figure 5. The peak in uncertainty at $\sim$730 cm$^{-1}$ can be attributed to the FORUM-aircraft apodisation described in Section 2.3.2.

temperature, gaseous and surface profile as in the FORUM-aircraft simulation (Section 2.3.2). The channels not used in the clear sky retrieval were first filtered based on whether clear sky residuals between LBLRTMv12.11 and RTTOVv12 were
comparable to, or exceeded the instrument uncertainty. This was to ensure that the uncertainty associated with the forward model could be directly attributed to its cloud assumptions. For each cloud property, these channels were then filtered based on two conditions. The first was that the largest changes in radiance were detectable above the measurement uncertainty. The second was that the radiance change was not perfectly correlated (i.e. a correlation coefficient of less than 1) to another channel. Channels with the largest change in radiance were selected first.
Figure 6 shows that 16 MIR and 10 FIR additional channels were selected to improve the retrieval of ice cloud properties. To highlight the channels that are most sensitive to ice crystal habit, Figure 7 shows the brightness temperature (BT) residuals within these channels between the GHM and SC models for five different CER (25, 30, 35, 40, and 45 µm) and a fixed IWP of $\sim$28 gm$^{-2}$. We see residuals between the models exceeding the measurement uncertainty between 400-450 cm$^{-1}$ (up to 1.3 K) and in the 1004.5 cm$^{-1}$ channel, and so expect the habit to be most detectable within these channels.

## 3.3 Assessing the Retrieval

In each set of retrievals, there are 4 different configurations used: with and without the FIR, and assuming the SC or GHM bulk optical property model. These will be referred to using the names outlined in Table 2 for clarity. Unlike the other retrieval targets, the ice crystal habit is retrieved by examining the spectral fit in certain channels after the optimal estimation retrieval of the other targets has been performed. To do this, we focus on the BT residuals in individual channels and the measurement cost per channel ($J_y$), which enables comparisons between different configurations. A lower $J_y$ for a given habit or habit mix is indicative that it is a better representation of the true crystal shapes within the cloud. If $J_y<1$, then the retrieval has fit the spectra on average within the uncertainty. For guidance, an increase of 20 % in $J_y$ is indicative of a 10 % increase in the residual between the observed and fitted spectra, assuming the uncertainty is kept constant.

The uncertainty in the retrieved state vector is calculated from the square-root-diagonal of the covariance of the retrieved state, $\mathbf{S}_x$:

$$\mathbf{S}_x = (\mathbf{S}_a^{-1} + \mathbf{K}^T\mathbf{S_y}^{-1}\mathbf{K})^{-1} \tag{8}$$

and is equivalent to one standard deviation. This will be referred to as the estimated standard deviation (ESD) in the retrieval.

The averaging kernel (AK) matrix, $\mathbf{A}$, represents the vertical sensitivity of the retrieved state to the true state, $\hat{\mathbf{x}}$ and is calculated using equation 9.

$$\mathbf{A} = \frac{\partial\hat{\mathbf{x}}}{\partial\mathbf{x}} = \mathbf{GK} \tag{9}$$

In this work, we assume the dropsonde profiles are a fair approximation to the true state, and they are smoothed to the resolution of the retrieval using:

$$\mathbf{x}_{AK} = \mathbf{x}_a + \mathbf{A}(\mathbf{x} - \mathbf{x}_a) \tag{10}$$

where $\mathbf{x}_{AK}$ is the AK-treated true state.

**Table 2.** The 4 different retrieval configurations used: with and without the FIR, and assuming the SC or GHM bulk optical property model.

| Name | MIR Channels | FIR Channels | Habit |
|---|---|---|---|
| MIR (GHM) | X | | GHM |
| MIR (SC) | X | | SC |
| MIR+FIR (GHM) | X | X | GHM |
| MIR+FIR (SC) | X | X | SC |

The degrees of freedom for signal (DOFS) is calculated from the trace of **A**, and is used to assess the amount of information in the measurement vector.

## 4    Fixed Temperature and Water Vapour Retrievals

Given the uncertainties associated with modelling the radiative interactions of ice clouds across the FIR, we perform two initial retrievals of COT, CER, and CTH, assuming a fixed temperature, gaseous, and surface profile. The temperature and water
vapour vertical profiles are fixed to the profiles measured by the dropsonde (Figure 2c and d), while the remaining gaseous and surface profiles are fixed to the values specified in Section 2.3.2. We only use the cloud-sensitive channels in this test as we are only focused on the cloud retrieval targets at this time. Therefore, when using the four retrieval configurations (Table 2), the MIR retrievals within this section only use 16 channels, and the MIR+FIR retrievals use all 26 channels shown in Figure 6.

The first set of retrievals are performed on the FORUM-aircraft simulation described in Section 2.3.2, and are used to assess
the impact of the fast scattering approximations used within RTTOVv12 (Section 4.1). The second set of retrievals are then performed on the FORUM-aircraft observation, where the ability of the bulk optical property models will influence the fitting process (Section 4.2).

### 4.1    Retrieval from the simulation

As described in Section 3, RTTOVv12 uses the Chou approximation to simulate cloudy radiances. This approximation intro-
duces a positive bias into the spectrum within the FIR that is dependent on the cloud parameters. Martinazzo et al. (2021) show that for an ice cloud of COT of 1, CER between 30-40 µm, and CTH of 8 km in a mid-latitude atmosphere, this positive bias peaks at $\sim$1K at $\sim$410 cm$^{-1}$. A lower COT and larger CER reduces this bias. Within the MIR, there is a negative bias that is approximately half the FORUM-aircraft uncertainty.

The FORUM-aircraft simulation described in Section 2.3.2 was simulated using the LBLDIS radiative transfer model, based
on the profiles in Figure 2 and assuming the GHM model. LBLDIS uses numerical methods to calculate the ice cloud interactions, and therefore by performing a retrieval on the FORUM-aircraft simulation, we can determine if the Chou approximation significantly hinders the retrieval capability as the inputs are known. The simulated spectrum has had Gaussian noise applied based on the instrument uncertainty (Figure 3) and includes the apodisation uncertainty associated with creating the FORUM-aircraft spectra.

Figure 8a shows the BT residuals associated with the retrieval from the FORUM-aircraft simulation with the four different retrieval configurations. When the correct (GHM) bulk optical property model and all 26 cloud-sensitive channels are used, the retrieved BT between 400-500 cm$^{-1}$ is, on average, within 0.2 K of the input simulation. Furthermore, the retrieved values in Table 3 capture the input parameters within two ESD. This suggests that despite the Chou approximation's bias in the far-infrared, we can assume that the retrieval scheme can reasonably estimate true values.

All four retrieval configurations also overestimate the BT in the channel at $\sim$730 cm$^{-1}$. This channel was selected due to its sensitivity to CER despite its increased uncertainty from the FORUM-aircraft apodisation (Figure 4a). The weaker fitting in

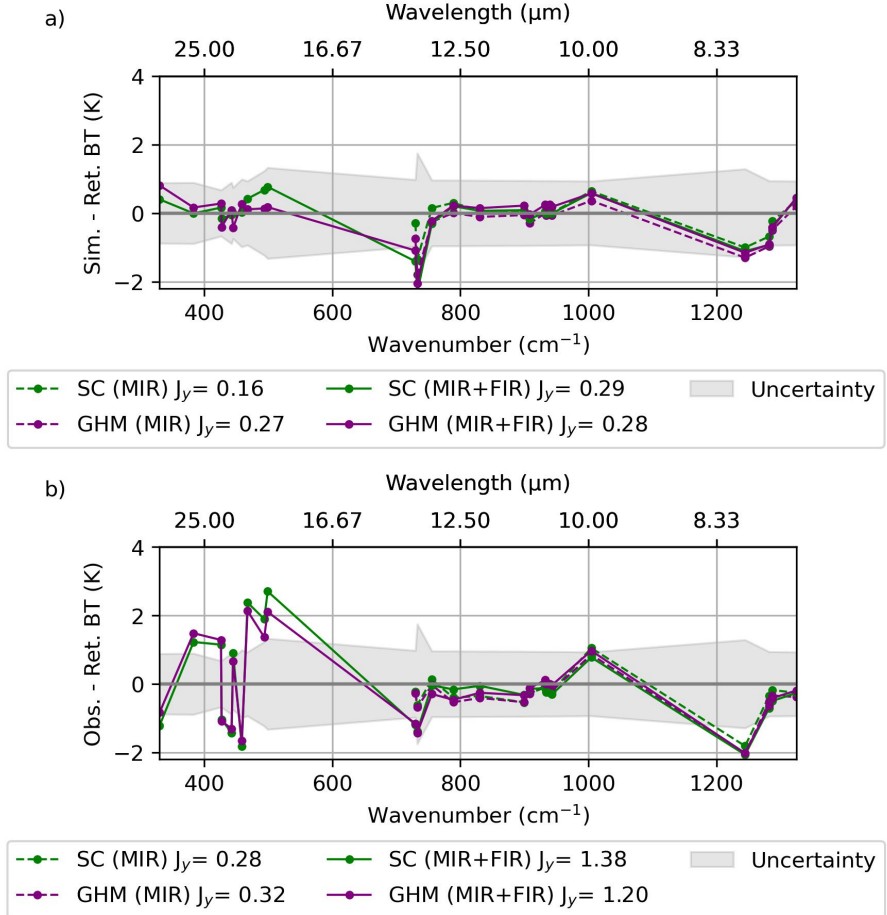

**Figure 8.** The BT residuals and $J_y$ in the cloud-sensitive channels using the (dashed) mid-infrared (MIR) only and (solid) mid- and far-infrared (MIR+FIR) on the FORUM-aircraft (a) simulation and (b) observation. The residuals within the MIR are partially obscured by the combined MIR+FIR retrieval. The retrievals using the SC and GHM models are shown in green and purple, respectively. The square-root-diagonal of $S_y$ is shaded in grey as the uncertainty.

this channel has likely contributed towards the larger CER ESD in Table 3, however, the input CER is still captured within one ESD for all the retrievals.

Given this, we can now examine the impact of including the FIR in retrievals from the simulation (Figure 8a). Using the MIR alone suggests that the SC model is a better representation of the ice crystal habit mix, with a measurement cost 40 % smaller than with the GHM model. This is an incorrect result as the GHM model was used in the simulation. When we include the FIR, the residual associated with the GHM model is up to 0.6 K smaller than its counterpart between 400-450 $\mathrm{cm}^{-1}$, comparable to expected residuals for CER in this range (Figure 7). This is indicative of a better fit, and correctly suggests the GHM model is the better representation of the ice crystals within the simulation. However, this is not significantly reflected in

**Table 3.** The retrieved COT, CER, and CTH from the FORUM-aircraft simulation described in Section 2.3.2 in comparison to the input values for the simulation. Uncertainties shown are one ESD. $J_y$ is also shown for each retrieval configuration and is the same as in Figure 8a.

|  | COT | CER ($\mu$m) | CTH (km) | $J_y$ |
|---|---|---|---|---|
| Input/True State | 0.82 | 34 | 9.0 | - |
| MIR (GHM) | 0.78±0.03 | 41±9 | 10.2±2.7 | 0.27 |
| MIR (SC) | 0.74±0.04 | 40±9 | 10.7±2.3 | 0.16 |
| MIR+FIR (GHM) | 0.84 ±0.04 | 38±8 | 9.5±0.3 | 0.28 |
| MIR+FIR (SC) | 0.82 ±0.04 | 35±8 | 9.1±0.3 | 0.29 |

the measurement cost, with only a slightly smaller $J_y$ for the GHM model. This is likely due to the larger BT residual for the GHM than SC habit in the $\sim$730 $\mathrm{cm^{-1}}$ channel countering the reduced residual in the FIR.

Table 3 shows that including the FIR leads to a closer agreement between all of the retrieved values and the known truth, as well as a significant reduction in the uncertainty associated with the CTH. The MIR+FIR GHM retrieval does show a slight positive bias in the cloud retrieval targets that is not seen with the MIR+FIR SC retrieval, as a result of reducing the

365 spectral residual caused by the Chou approximation. However, it is important to recall that the retrieval is assessed using the measurement cost and spectral fit, and that the MIR+FIR GHM retrieval still captures the true state within 2 ESD.

### 4.2 Initial retrieval from the observation

As in the previous section, we perform retrievals of COT, CER, CTH, and ice crystal habit with a fixed vertical profile, but now on the radiance observation. Given that the fixed vertical profile is now only an approximation of the true state, we will

not explore all the retrieval targets in detail as these will be discussed in the context of the full retrieval in Section 5. In this section, we only focus on the retrieved ice crystal habit and BT residuals to help separate the influences of the vertical profile and cloud approximations on the full retrieval in Section 5.

The in-situ measurements of the ice crystal habits suggest that the GHM model is a better representation of the ice crystals within the cloud than the SC model (Section 2.2). Figure 8b shows the BT residuals for retrievals from the FORUM-aircraft

observation using only the cloud-sensitive channels and we observe similar trends to the simulated case. The measurement cost per channel using the MIR alone is again slightly larger for the GHM model than the SC model, with retrieved spectra within 0.2 K of each other. When we introduce the FIR channels, $J_y$ is reduced by 13 % using the GHM model. This reduction is driven by the fitting between 450-500 $\mathrm{cm^{-1}}$, where there are differences of up to 0.6 K between the MIR+FIR SC and GHM retrievals. In common with the simulated results from Section 4.1, this strongly implies that for this case study, observations

from the FIR are needed to correctly diagnose the ice crystal habit.

However, the BT residuals within the FIR generally exceed the measurement uncertainty, particularly between 450-500 $\mathrm{cm}^{-1}$, while there is little change in the MIR residuals with and without the FIR. This was not seen in the retrieval from the FORUM-aircraft simulation, which used the same bulk optical property model as the retrieval scheme. This reaffirms the findings in Bantges et al. (2020), which highlighted the inability of bulk optical property models to simultaneously match observed radiance signals in the MIR and FIR. Despite this, the bulk optical property models do not prevent the inclusion of the FIR from distinguishing between ice crystal habits.

It is important to note that the temperature and water vapour profile are currently fixed to measurements from the dropsonde. To test that the fixed temperature and water vapour profile is not biasing the habit determination, we now also include the simultaneous retrievals of temperature, and water vapour from the radiance observation.

## 5 Retrieval of Temperature, Water Vapour, and Cloud Properties

Retrievals of temperature, water vapour, COT, CER, CTH, and ice crystal habit are performed using both the clear sky and cloud-sensitive channels shown in Figure 9 on the FORUM-aircraft observation. The same four configurations described in Table 2 are used, however the MIR retrievals now use 116 channels, and the MIR+FIR retrievals use 226 channels (Figure 6). In each case, the a-priori and a-priori covariance for temperature, water vapour, COT, CER, and CTH are as described in Section 3.

We divide the following discussion into three subsections: Section 5.1 focuses on the retrieval of the ice crystal habit, Section 5.2 explores the influence of ice crystal habit and the FIR on the retrieved temperature and water vapour profiles, and Section 5.3 evaluates the retrieved cloud properties against the in-situ measurements described in Section 2.2.

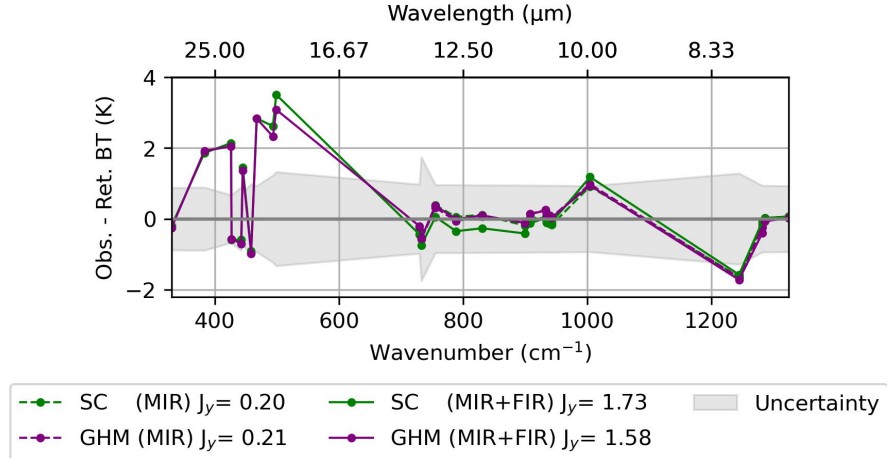

**Figure 9.** The BT residuals and $J_y$ for retrievals from the observation. All 226 channels are used in the retrieval, but only residuals and $J_y$ in the 26 cloud sensitive channels are shown for comparison to Figure 8b. $J_y$ across all 116 (MIR) and 226 (MIR+FIR) channels is as follows: SC (MIR) 0.33, GHM (MIR) 0.34, SC (MIR+FIR) 1.20, and GHM (MIR+FIR) 1.15.

## 5.1 Retrieval of the Ice Crystal Habit

As discussed in Section 2.2, the in-situ observations of the ice crystals suggest that the GHM model is a better representation of the ice crystals within the cloud. Figure 9 shows that using the MIR alone does not allow a clear distinction between the GHM and SC models when we only consider the spectral fit. In the cloud-sensitive channels, we see comparable values of $J_y$ for the MIR GHM and MIR SC configurations, similar to the findings in Section 4.2. This is also true across all 116 MIR channels which include greater spectral sensitivities to the temperature and water vapour vertical profile. This suggests that while the inclusion of temperature and water vapour in the retrieval does affect the BT residuals, it does not significantly impact our inability to distinguish between habits when we only use the MIR.

When we include the FIR, there is a clearer distinction between habit models than when we only use the MIR, with the $J_y$ for MIR+FIR GHM configuration 9 % smaller than its SC counterpart in the cloud-sensitive channels. This smaller $J_y$ for the MIR+FIR GHM configuration suggests that the GHM model enables a better spectral fit, and so is the more likely representation of the ice crystals within the cloud. This is in line with the in-situ measurements.

The addition of temperature and water vapour in the MIR+FIR retrieval has a more notable impact on the spectral distinction between ice crystal habits than when we only use the MIR channels. This is most clearly seen when we compare the BT residuals associated with the MIR+FIR SC and GHM configurations within the FIR (Figure 9). Below 410 $cm^{-1}$, the BT residuals between the GHM and SC configurations are now comparable. Between 410 and 600 $cm^{-1}$ the residuals differ by, on average, 0.1 K, which is 0.5 K smaller than when the vertical temperature and water vapour profiles were fixed to the dropsonde measurements (Figure 8b). This similarity is reflected by the reduction in percentage difference between $J_y$ in the cloud-sensitive channels for the MIR+FIR GHM and SC configurations by 6 % relative to the retrievals with the fixed vertical profile. Furthermore, when we consider $J_y$ across the clear-sky and cloud-sensitive channels, the percentage difference between $J_y$ for the MIR+FIR SC and GHM configurations is only 5 %. This effect is a result of the differences in the associated retrieved temperature and water vapour profiles and will be explored further in Section 5.2.

## 5.2 Retrieval of Temperature and Water Vapour

The benefits of including the FIR in clear sky retrievals of upper tropospheric water vapour has been explored on observations in mid-latitude conditions (Warwick et al., 2022; Panditharatne et al., 2025). Our results indicate that including the FIR increases the DOFS for temperature and water vapour from 2.9 to 3.8 and from 1.9 to 3.7, respectively, due to the additional information contained within the FIR OLR spectrum. Both the MIR and MIR+FIR GHM retrievals generally capture the AK-treated dropsonde profile within 2 ESD (not shown). However, as previously discussed (Section 5.1), including temperature and water vapour in the retrieval influences the determination of the ice crystal habit for the retrievals that include the FIR. This is in part due to the increased sensitivity of FIR radiances to the water vapour and temperature profile as reflected in the DOFS.

Figure 10 shows the retrieved temperature and water vapour profiles using all 226 channels in the MIR+FIR GHM and SC configurations. The temperature retrieval for both habit models is worse than the a-priori with residuals of up to 1 K from the AK-treated dropsonde temperature profile (Figure 10b). This is in part due to the increased uncertainty in the 15 micron

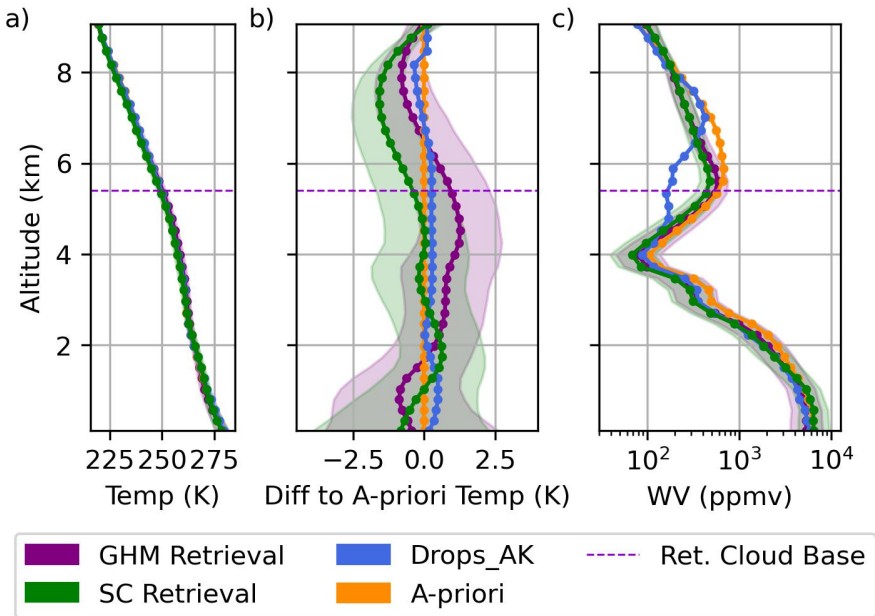

**Figure 10.** The retrieved (a) temperature and (c) water vapour profile from the FORUM-aircraft observation with (b) differences between the retrieved and AK-treated dropsonde temperature profiles. The retrievals assuming the General Habit Mix (GHM) and Solid Columns (SC) models are shown in purple and green, respectively, and use all 226 channels. The average cloud base from the two retrievals is shown with the dashed purple line. The shaded region around each retrieval represents one ESD. The a-priori is shown in orange, with the surrounding shading showing one standard deviation. The AK-treated dropsonde (Drops_AK) profiles are shown in blue. The uncertainty in the dropsonde measurements are discussed in Section 2 but are not visible on the scale of this figure. For temperature, the dropsonde uncertainty is 0.2 K. For water vapour, the dropsonde uncertainty varies with altitude but remains below 9%.

$CO_2$ wings following the FORUM-aircraft apodisation (Panditharatne et al., 2025). Despite this, the AK-treated dropsonde temperature profile is within one ESD of both retrievals below 6 km, and entirely within the GHM retrieval. For the MIR+FIR case, assuming the SC habit reduces the retrieved temperature between 2-8.5 km relative to the GHM case, with differences
reaching up to 1.9 K. A reduction is also seen between the SC and GHM cases if only MIR channels are included, but in this case it is smaller, reaching a maximum of 0.4 K (not shown). This reduction in temperature lowers the simulated brightness temperature throughout the channels, particularly between 450-500 $cm^{-1}$ to balance the changes in the cloud retrieval products that will be discussed in Section 5.3.

Figure 10c shows that the habit does not affect the retrieved water profile outside of one ESD, with the median retrieval bias
throughout the vertical only increasing by 2 % from the GHM to SC model. Both retrievals capture the AK-treated dropsonde profile below 4 km and above 8 km, but struggle at the base of the cloud. Neither retrieval captures the observed dry patch around the cloud base, as they remain close to the a-priori.

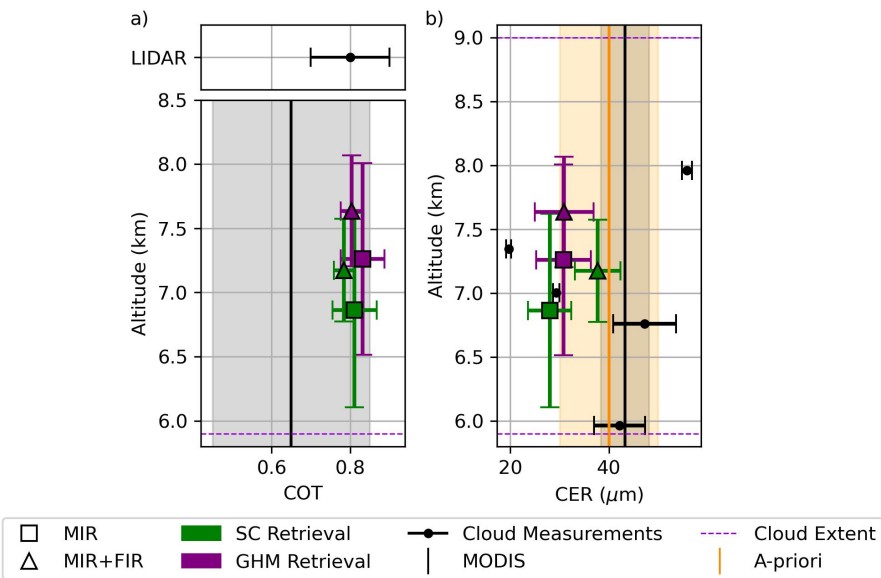

**Figure 11.** The retrieved (a) COT and (b) CER from the FORUM-aircraft observation in comparison to measurements of the cloud. The IMS retrieved values are shown using squares (116 MIR channels) and triangles (226 MIR+FIR channels). The retrievals using Solid Column only (SC) and the General Habit Mix (GHM) are shown in green and purple, respectively. Error bars show one ESD in the retrieved CTH, COT, or CER. The measurements of the cloud taken by instruments on board the plane are shown as black dots. For the COT, we show the mean value derived from lidar measurements between 09:48:09 to 09:49:09 UTC, with the error bar representing the standard error of the mean. For the CER, the black dots are calculated from the PSDs measured by the Cloud 2-DS probe at 5 altitudes within the cloud, and the error bars show one standard deviation derived from the bimodal fit. The vertical black line shows the median MODIS L2 COT and CER from the 10:40 UTC overpass between 58.7-59.75°N and 4.2-3.3°W at 1 km resolution. The shading around the MODIS COT and CER is one median absolute deviation and shows the spread of values within this spatial region. The a-priori for the CER is shown in orange with shading representing one standard deviation. The a-priori for the COT is 0.01 and is not shown.

## 5.3 Retrieval of COT, CER, and CTH

Figure 11 shows the retrieved values for the COT and CER in comparison to the measurements of the cloud described in Section
2.2. The retrieved values are plotted at the midpoint of the retrieved cloud to allow a comparison to the individual CERs derived from the PSDs. While the lidar-derived COT was taken 4 seconds after the radiance observation, the in-situ measurements of the cloud were taken between 10:32 and 11:32 UTC, and so we also include the L2 COT and CER from the MODIS Terra overpass at 10:40 UTC. The MODIS data have been averaged across the spatial range of the in-situ measurements and radiance observation (58.7-59.75°N and 4.2-3.3°W).

There is some evidence of systematic behaviour between the COT retrievals shown in Figure 11a. For example, the GHM retrievals (purple points) show slightly larger COTs than their corresponding SC retrievals. Similarly, the MIR retrievals (square points) show slightly larger COTs than the MIR+FIR retrievals. However, all 4 configurations retrieve the COT within the

uncertainty of the lidar-derived COT of 0.8±0.1 that was colocated with the radiance observation, and are within one median absolute deviation of the median MODIS COT taken approximately an hour after (Figure 11a). The relative insensitivity of the COT retrieval to the inclusion of the FIR is expected given the strong sensitivity to COT across most of the MIR channels used in the retrieval (Bantges et al., 2020), and this is reflected in the minimal increase in the DOFS from 0.9 to 1.0 when the FIR is included.

The centre of the cloud measured by the lidar is at ∼7.5 km, and we see this captured within the range of all 4 retrieval configurations, with a consistent DOFS of 1 for CTH across all the retrievals. The inclusion of the FIR (triangle points) slightly shifts the cloud higher by ∼0.4 km, but the retrievals still remain within one ESD of their MIR counterparts (square points). In contrast, assuming the SC model rather than the GHM model can be seen to slightly lower the retrieved cloud by ∼0.4 km, and this is likely to counteract the colder vertical profile that is retrieved with this habit (Figure 10).

For both the retrieved CTH and COT, we see the ESD halve from 0.8 to 0.4 km and 0.06 to 0.03, respectively, with the inclusion of the FIR. This suggests a greater confidence in the state and agrees with previous simulation studies that suggest including the FIR could reduce uncertainties in cloud retrieval targets (Libois and Blanchet, 2017).

All 4 retrievals of the CER are between 28 and 38 μm. This is generally lower than the a-priori and MODIS CER, and tends towards the smaller in-situ measurements. There is a notable difference in sensitivity to the inclusion of FIR information depending on the habit selected in the retrieval. For the GHM case the retrieved CER is unchanged at 31±6 μm regardless of whether FIR channels are included, showing good agreement with the weighted mean of the in-situ CER measurements (32±14 μm). In contrast, if the SC habit is selected, the retrieved CER increases from 28±4 to 38±5 μm from the MIR only to the MIR+FIR case, with the MIR+FIR SC retrieval remaining closest to the a-priori. There is also a slight increase in the CER DOFS from 0.7 to 1.0 when the FIR is included.

Overall, despite the limitations associated with the bulk optical property models, and the use of a fast cloud scattering approximation we are able to retrieve cloud properties that agree with the independent lidar and in-situ observations. The addition of FIR information provides a greater constraint on the inferred COT and CTH (reduced retrieval uncertainty) and, importantly, allows an improved discrimination of crystal habit compared to retrievals using only the MIR part of the spectrum.

## 6 Conclusions

We present the first retrievals of cirrus cloud properties (optical thickness, effective radius, cloud position, habit), temperature, and water vapour that exploit upwelling mid- and far-infrared radiances (100-1600 $cm^{-1}$) observed from an aircraft. These airborne radiances have been adapted to emulate the FORUM Sounding Instrument's spectral characteristics to provide an indication of the benefit that the FORUM satellite mission could bring to our current understanding of ice clouds.

We make use of the RAL Infrared Microwave Sounding (IMS) retrieval scheme which we have modified to encompass different cloud optical property models within the far-infrared. This retrieval scheme employs the optimal estimation approach and uses the Chou approximation to simulate cloudy radiances. While this fast scattering approximation is known to introduce

a spectral bias within the far-infrared, sensitivity tests showed that this does not compromise the ability of IMS to retrieve the 'true' state for this case.

Our primary aim was to investigate whether the inclusion of the far-infrared improved the quality of the retrieved targets, with a particular focus on the ice crystal habit. To do this, four different retrievals were performed on the observation: with and without the far-infrared, and with either the Solid Column or General Habit Mix bulk optical property model (Yang et al.,
2013; Baum et al., 2014). The outputs were then evaluated against colocated measurements from the on-board lidar and in-situ measurements of the ice crystal habits and PSDs that were taken an hour after the radiance observation.

The retrieved cloud optical depth and position of the cloud were consistent with the lidar measurements for all four configurations. However, we find that the inclusion of the far-infrared halved the retrieval uncertainties for these retrieval targets in comparison to when only the mid-infrared was used. These results are in line with previous findings based on theoretical
simulations and ground-based observations (Libois and Blanchet, 2017; Di Natale et al., 2017).

Our results also demonstrate that including the far-infrared enabled a clearer and more likely distinction of the ice crystal habit for this case. When only the mid-infrared was used, the spectral fit suggested that the Solid Column model was a slightly better representation of the ice crystals within the cloud, which conflicted with the in-situ measurements. The addition of the far-infrared resulted in a better spectral fit for the General Habit Mix which was more compatible with the inferred habit
distributions. Furthermore, the choice of habit had a significant influence on the retrieval of both temperature and the cloud effective radius, with stronger deviations observed when the far-infrared was included.

We note that despite these encouraging results, there are caveats associated with this study. The in-situ measurements that we use to derive the cloud effective radius were taken an hour after the radiance measurements and it is possible they may have evolved with time. Perhaps more critically, the spectral residuals obtained after the retrievals have converged exceed the
measurement uncertainty within the far-infrared. While our results imply that this does not preclude using the selected (and widely used) ice optical property models chosen here, it does imply that they need refinement to be consistent across the entire infrared. This finding is in full agreement with previous work and motivates both the further development of these models, and the need for a larger set of coherent observations of in-situ cloud properties and spectral radiances across the entire infrared.

*Data availability.*  The ERA5 data are from the Copernicus Climate Change Service (C3S) Climate Data Store Copernicus Climate Change
Service (2018). Data from the CIRCCREX field campaign can be found at the Natural Environment Research Council's Data Repository for Atmospheric Science and Earth Observation: https: //catalogue.ceda.ac.uk/uuid/6ba397d6c8854da19bcced8ea588c1f9 (CEDA, 2016). The MODIS data is taken from Platnick et al. (2003).

*Author contributions.*  The study design was conceived by SP, HEB, and CC. SP optimised the retrieval framework, performed the retrievals, and wrote the first draft of the manuscript. HEB, and CC provided expertise, contributed to discussions of the results and revisions of the
manuscript. RS (University of Oxford) developed the technique to infer the COT from the lidar measurements. RS (RAL Space) built the

retrieval framework. RB provided expertise on the radiative simulations and the B895 flight. JEM and CF took the measurements from the TAFTS instrument as part of the CIRCCREX campaign. SF provided information about the ARIES measurements and lidar data.

*Competing interests.* The authors declare that they have no conflict of interest.

*Acknowledgements.* Sanjeevani Panditharatne was funded by NERC (under grant no. NE/S007415/1) via the SSCP DTP. The CIRCCREX
campaign and processing of these data were funded by the Natural Environment Research Council (UK) grant nos. NE/K015133/1 and
NE/K01515X/1. Airborne data were obtained using the BAe 146-301 Atmospheric Research Aircraft flown by Airtask Ltd and managed by
FAAM Airborne Laboratory, now jointly operated by UKRI and the University of Leeds. The authors would like to thank the instrument
operators, aircrew, operations staff, and engineers for their support during the campaign.

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
