# Peer review of "Exploiting airborne far-infrared measurements to optimise an ice cloud retrieval"

_EGUsphere, 2025_

## Referee Comment (RC1)

**Review of "Exploiting airborne far-infrared measurements to optimise an ice cloud retrieval" by Panditharatne et al.**

**General comments**

Retrievals using the far- and mid-infrared have been carried out using synthetic data (e.g. Saito et al. 2020). This study provides the first retrievals (to the knowledge of the authors and this reviewer) of ice cloud properties using an observation of coincident upwelling far- and mid-infrared radiances taken during the CIRCCREX campaign. Since the study is in the frame of the FORUM mission, the observed radiances are adapted to mimic what would be observed by FIS (Forum Sounding Instrument), the so called FORUM-aircraft observations.

This work aims to explore if adding the far-infrared benefits the retrievals, especially regarding the distinction of ice crystal habits, and to assess the limitations of two bulk cloud optical property models (Solid Column and General Habit Mix) on the retrievals. The authors perform simultaneous retrievals of cloud optical thickness, cloud effective radius, cloud top height, temperature and water vapor and compare the results to in-situ measurements.

The current study expands the work done in Panditharatne et al. 2025, which describes the method to create the FORUM-aircraft observations and the retrieval of temperature and water vapor. Comments regarding this methodology have been assessed by the reviewers of the mentioned paper.

In general, I find it an interesting article that is well structured, although there are a few points that get confusing (see comments below). I consider it can be published after assessing the following specific and technical comments:

**Specific comments**

Line 126: what bimodal function has been fitted to the PSDs?

Line 127: is the habit of particles smaller than 50 um. assumed spherical? What is the range of sizes from the in-situ measurements?

Line 128: is the distribution of habits similar for each of the sampled altitudes or are there significant differences?

Line 150: how is the quality of the fit of the PSD evaluated?

Line 152: "some work suggesting that the smaller mode of the distribution is an artefact of the instrumentation" If the bimodality of the PSDs is real or an artefact is indeed still a matter of debate. There is a lot of work done to correct and avoid the shattering of ice particles to provide a database of PSDs of ice crystals where bimodality is not associated to shattering. I would suggest having a look at, e.g. Krämer et al., 2020 and references therein.

Line 154: can you explain why the uncertainty in CER is larger the closer to the bottom of the cloud?

Line 173: why were 6 scans chosen?

Line 184: "… the simulated FSI and FORUM-aircraft spectra produced similar results, suggesting a retrieval from an FORUM-aircraft observation is indicative of its FSI counterpart given a homogeneous scene". I would recommend to add a few sentences explaining why the selected cloud is homogeneous and therefore the method is applicable to the studied case.

Figure 7: any comments on the pick of the uncertainty at around 750 cm$^{-1}$ and the BT residual for 35 um being consistently lower than for the rest of CRE?

Line 325: "When the correct (GHM) bulk optical... ", what do you mean here by correct?

Line 335: "This is an incorrect result as the GHM model was used in the simulation". I don't understand this sentence.

Line 333-342: here it is discussed that the results using the GHM model are better, however, when looking at Table 3, MIR+FIR (SC) is the combination that delivers the closest results to the considered true state. I would suggest expand the discussion about GHM vs SC.

Line 357-360: I would add a few lines about why the observations exceed the uncertainty but not the simulation.

**Technical comments**

Line 79: this comment is a little bit picky, but what do you understand as a well characterized cirrus cloud?

Figure 1: what is the difference between the continuous blue line and the dashed blue lines? If the first line was changed to continuous because of overlapping with the cloud extent, I would suggest marking the extent of the cloud with a continuous line, so all the in-situ are dashed and picking two colors that contrast a little bit more than blue and purple.

Line 33: add reference to some of the studies.

Line 75: add reference to the website of CIRCCREX.

Figure 2: I would add in the caption the altitude of the dashed blue lines so it is faster to identify which line corresponds to the the altitudes specified in Table 1. "… have been characterised are shown in blue (6, 6.8, 7, 7.3, 7.5, 8., 8.3 km)"

Table 1: As a suggestion, I would add another column with the temperature.

Line 144: number distribution of sizes → size number distribution (?)

Line 147: I would recommend adding some references regarding the bimodality of cirrus clouds. For temperatures larger than approx. 210 K, bimodality starts playing a role. For lower temperatures, a fit to a monomodal PSD should be expected.

Line 179: for consistency, I would give the FOV of ARIES, TAFTS and FIS in the same units.

Line 180: although the acronyms of NEDT and ARA appear in the caption of Fig. 3, I would suggest to write 'The apodised noise-equivalent-differential-temperature (NEDT) and target absolute radiometric accuracy (ARA)'.

Line 205: here is mentioned for the first time the "FORUM-aircraft simulation", but it passes a little bit unnoticed. For the rest of the manuscript it gets a bit confusing what the FORUM-aircraft observation is and what the FORUM-aircraft simulation is. I would suggest to rephrase this line to make clearer what the differences between the two are. Also in the caption of Fig. 4 is written "FORUM-aircraft LBLDIS simulation", is this the same as FORUM-aircraft simulation? Same for caption in Fig. 8 and caption in Table 3 and in line 310 where is written "LBLDIS GHM FORUM-aircraft simulation".

Line 216: … and in the presence of cloud → … and in the presence of clouds (?)

Line 245: "the ice cloud is modelled as a Gaussian of 1 km thickness". Is 1 km at full-width-half-maximum?

Line 247, eq. 7: isn't a dz missing? And are the limits from 0 to z or from CBH to CTH? Since the ice crystal is much larger than the wavelength, is $<Q_{e,vis}>$ assumed to be 2?

Line 278: for convenience for the reader, I would add also here the five CER and the number for the fixed IWP.

Figure 7: in the caption I would specify again of what the black line is uncertainty of.

Line 372: "…, before Section 5.3 evaluates the …" → "… and Section 5.3…"

Line 379: verticle → vertical

Line 399: for those not familiar with the term, I would specify "DOFS (degrees of freedom)".

Caption of Fig. 9, lines 2-3: I would specify Figure 8b. Also I would add ":" after "...as follows" and I would change the "." for "," in between SC(MIR): 0.33**.** GMH()...

Line 423: These have → It has (?)

Line 431: do you mean from 0.9 to 1?

Caption of Fig. 11, in line 4: '… or CER. The error bars The measurements of the cloud taken…' → delete 'The error bars'. In line 5: … from lidar measurements between from 09:48:09 to 09:49:09 → delete 'between' or delete 'from' and change 'to' for 'and'.

Line 454: "… 1600 cm-1" → cm$^{-1}$

Line 484: https://doi.org/10.24381/cds.bd0915 doesn't work.

Line 487: "The MODIS data is taken from" this sentence is not finished.

Line 528: https://doi.org/https://doi.org/10.1016/j.jqsrt.2022 – delete the first https://doi.org/

Line 530: '...Cycle Coopera-tive' → '...Cycle Cooperative'

Line 548: https://doi.org/https://doi.org/10.1029/2022GL099394, e2022GL099394 2022GL099394, 2022 → https://doi.org/10.1029/2022GL099394, 2022

Line 564: I don't find the technical report.

Line 601: https://doi.org/https://doi.org/10.1029/2021JD035733, e2021JD035733 2021JD035733, 2022 → https://doi.org/10.1029/2021JD035733, 2022

Line 631: add doi: https://doi.org/10.1175/JAMC-D-11-067.1

Line 638-639: Delete " and add doi: https://doi.org/10.1175/JAS-D-12-039.1

**References**

Krämer, M., Rolf, C., Spelten, N., Afchine, A., Fahey, D., Jensen, E., Khaykin, S., Kuhn, T., Lawson, P., Lykov, A., Pan, L. L., Riese, M., Rollins, A., Stroh, F., Thornberry, T., Wolf, V., Woods, S., Spichtinger, P., Quaas, J., and Sourdeval, O.: A microphysics guide to cirrus – Part 2: Climatologies of clouds and humidity from observations, Atmos. Chem. Phys., 20, 12569–12608, https://doi.org/10.5194/acp-20-12569-2020, 2020.

Panditharatne, S., Brindley, H., Cox, C., Siddans, R., Murray, J., Warwick, L., and Fox, S.: Retrievals of water vapour and temperature exploiting the far-infrared: application to aircraft observations in preparation for the FORUM mission, Atmos. Meas. Tech., 18, 717–735, https://doi.org/10.5194/amt-18-717-2025, 2025.

Saito, M., Yang, P., Huang, X., Brindley, H. E., Mlynczak, M. G., & Kahn, B. H. (2020). Spaceborne middle- and far-infrared observations improving nighttime ice cloud property retrievals. *Geophysical Research Letters*, 47, e2020GL087491. https://doi.org/10.1029/2020GL087491

---

## Referee Comment (RC2)

**Exploiting airborne far-infrared measurements to optimise an ice cloud retrieval  by Panditharatne et al.**

This is an interesting and valuable work. It describes the first retrieval of the cloud  parameters, including optical depth, effective radius and top height, along with the atmospheric profiles of temperature and water vapour from up-welling far-infrared spectral radiances. In this work the RAL retrieval scheme, introduced in previous works, has been applied to the spectral radiance measurements performed with two spectrometers: the Tropospheric Airborne Fourier Tranform Spectrometer (TAFTS), operating in the far infrared (FIR) portion of the spectrum between 80-600 cm-1, and the Airborne Research Interferometer Evaluation System (ARIES) operating in the mid infrared (MIR) between 550-3000 cm-1. These measuremenets were performed on board of the aircraft B895 which flew in 2015 as part of CIRCCREX campaign. Carrying on board several  instruments for cloud and atmosphere characterization, such as a series  of three probes or CPIs and a backscattering/depolarization lidar to assess the cloud particle habit/size distributions and the extinction coefficient along with Vaisala probes providing information about humidity and temperature. The availability of such in situ measurements allowed the comaparison with the retrieval products. The results show in generally a good accordance but also point out on the necessity of a refinenement of the currrent ice optical properties in models particularly in the FIR. Furthermore, it is shown the improvement in the retrieval performance by using the FIR portion  with respect to neglect it, in particular in distinguishing the different crystal habits.

The manuscript is well written and structured even though some minor revisions need to be addressed before publication. I also suggest few corrections to enhance the clarity:

Abstract:  line 5  “:with and without the far infrared..” → “including and neglecting the far infrared portion of the spectrum..”

line 17:  “..cover..” → “..cover permanently..”

line 54: “This work” → “The present work..”

Section 2.2

Please, can you structure the initial part of the section by listing the in situ instruments present on board for cloud characterization, describing briefly what

they provide? I think it is more simple for a reader to follow the text having the scheme in mind.

You say that the extinction coefficient profile obtained from lidar signal in Fig. 2a is not reliable for deriving optical depth due to calibration issues; if you don't use it to calculate the COT, maybe, this is a little bit misleading, wouldn't be better to show the raw signal?

Regarding this, the COT calculated in Eq. (1) with the new method, is it obtained using the raw signal? I mean, are the Patt and Pref raw signals? Maybe it is better to report this in the text. If so, this is a reason to show the raw signal. in Fig. 2.

However, how much the COT differ from that one calculated from the extinction profile? By eye, it does not look too far from the value you found, if we multiply the thickness for the mean extinction, but maybe I'm wrong. How the extinction coefficient has been derived from the backscattering signal?

The extinction due to the molecular contribution is already considered in the new method?

Caption Fig. 2: please indicate that are dashed lines.

Fig. 2 : the blue lines shouldn't be dashed?

I would suggest , if possible, to show some pictures of the habit and size distributions provided by the CPIs. It is just a suggestion because it would be very interesting to see the in situ measuremenets of the ice crystals inside cirrus. And of course, I think, it would be an added value.

Section 2.2.1

Line 145: can you indicate the size range of the dominant smaller crystal?

Can you show which parameters are fitted of the PSDs and report some results of the fit? Or briefly explain the procedure?

Line 148: "..the habit and.." → "..the habit type and.."

Line 148: "..CPI measurements habit recongnition.." → "..CPI measurements by applying habit recongnition.."

Can you report, if possible, a numerical example of the habit fractions derived from CPIs by applying the recognitional algorithm?

**Section 2.3.1**

Line 186: with "remove" you mean deconvolved?

Line 191 and 193: Just remember that are available more recent version of LBLRTM and, in particular, of continuum MT_CKDv3.8

Line 199: "..using the Masuda model above 769cm-1" → "..using the Masuda model above 769cm-1 since the radiometric measurements were performed over the Ocean" (Is it correct? If so I would clarify why the Masuda model has been used)

Line 203: "and the GHM model" → "assuming the GHM model"

Fig. 4: The differences of TAFTS in blue are not visible, would be possible to expand the scale just for this plot of TAFTS for example in logarithmic scale to make it more readable?

**Section 3**

Could you indicate what is the initial guesses or if they are equal to a-priori?

Which are the correlation lengths used for a-priori atmospheric profiles?

Eq. (7) the differential dz is missing inside the integral.

If you fit the CTH how much did you fix the geometrical thickness?

Line 431: maybe you mean "..increase from 0.9 to 1.0"? not from 0.9 to 0.1

---

## Author Response (AR1)

**RC1 Comments**

**Review of "Exploiting airborne far-infrared measurements to optimise an ice cloud retrieval" by Panditharatne et al.**

**General comments**

Retrievals using the far- and mid-infrared have been carried out using synthetic data (e.g. Saito et al. 2020). This study provides the first retrievals (to the knowledge of the authors and this reviewer) of ice cloud properties using an observation of coincident upwelling far- and mid-infrared radiances taken during the CIRCCREX campaign. Since the study is in the frame of the FORUM mission, the observed radiances are adapted to mimic what would be observed by FSI (Forum Sounding Instrument), the so called FORUM-aircraft observations.

This work aims to explore if adding the far-infrared benefits the retrievals, especially regarding the distinction of ice crystal habits, and to assess the limitations of two bulk cloud optical property models (Solid Column and General Habit Mix) on the retrievals. The authors perform simultaneous retrievals of cloud optical thickness, cloud effective radius, cloud top height, temperature and water vapor and compare the results to in-situ measurements.

The current study expands the work done in Panditharatne et al. 2025, which describes the method to create the FORUM-aircraft observations and the retrieval of temperature and water vapor. Comments regarding this methodology have been assessed by the reviewers of the mentioned paper.

In general, I find it an interesting article that is well structured, although there are a few points that get confusing (see comments below). I consider it can be published after assessing the following specific and technical comments:

We thank the reviewer for their overall positive review, comments and recommendations. A number of changes have been made to the paper to improve clarity. We have particularly focused on descriptions relating to the in-situ measurements of the PSDs, and the discussion surrounding the FORUM-aircraft simulation.

**Specific comments**

Line 126: what bimodal function has been fitted to the PSDs?

Gaussian bimodal – added.

Line 127: is the habit of particles smaller than 50 um. assumed spherical? What is the range of sizes from the in-situ measurements?

In O'Shea et al. (2016), particles smaller than 50 microns are not used in the habit determination. In this work, it is assumed that particles smaller than 50 microns are not spherical and have ice crystal habits that can be modelled using the General Habit Mix.

This sentence has been rewritten for clarity:

*The ice crystal habits within the cloud were determined from CPI images of crystals larger than 50 microns using an automatic habit recognition algorithm outlined in O'Shea et al 2016.*

Line 128: is the distribution of habits similar for each of the sampled altitudes or are there significant differences?

Added:

*Particles were found to be a mix of predominantly aggregates (~30 %) and droxtals (~25 %), with some rosettes (~15 %) and columns (~10 %) throughout the cloud.*

Line 150: how is the quality of the fit of the PSD evaluated?

The original phrasing has been replaced with:

*The uncertainty presented here for the CER is calculated using the relative standard deviation of the fitting parameters and so represents one standard deviation in the fitting of the bimodal Gaussian distribution to the PSD.*

The fitted PSDs and a table of the relative standard deviation in each value has also been included in the Supplementary information (Figure S1 and Table S1).

Line 152: "some work suggesting that the smaller mode of the distribution is an artefact of the instrumentation" If the bimodality of the PSDs is real or an artefact is indeed still a matter of debate. There is a lot of work done to correct and avoid the shattering of ice particles to provide a database of PSDs of ice crystals where bimodality is not associated to shattering. I would suggest having a look at, e.g. Krämer et al., 2020 and references therein.

It is unlikely that the smaller mode of the PSDs in this work is due to shattering of the ice crystals. The probes were fitted with anti-shatter tips to prevent ice breakup on their leading edges. Furthermore, the crystals were filtered based on the interarrival times (described in O'Shea et al, 2016) and it was found that the 'shattering mode' could be separated from the data, removing the effect of shattered particles on the distributions used in this work.

The artefact of the instrumentation referenced here is outlined in the study cited on Line 152 (O'Shea et al, 2021). This study analysed the PSDs measured during another CIRCCREX flight that used the same instrumentation as the B895 flight. This study found that a diffraction pattern surrounding the 'real' ice crystals was observed that could contribute to the smaller mode in the PSDs. The authors use a similar method (as is used for the PSDs analysed in this work) of filtering by interarrival times between the two stereo probes to reduce this smaller mode. However, there are still artefacts observed in the PSDs.

This sentence has been adjusted for clarity:

*We note that larger uncertainties are expected in the PSDs themselves, with some work suggesting that the smaller mode of the distribution is linked to diffraction effects from the instrumentation (O'Shea et al., 2021).*

Line 154: can you explain why the uncertainty in CER is larger the closer to the bottom of the cloud?

The uncertainty in the CER is derived from the fitting of the bimodal Gaussian to the PSDs measured by the Cloud 2-DS probe. There is a larger number of 'mid-size' crystals at the lower altitudes (see Figure S1 in the Supplement) which weakens the fitting and results in a larger uncertainty.

Line 173: why were 6 scans chosen?

Six scans were chosen to equate ARIES (0.25s) and TAFTS (1.5s) scan times. A correction was made to line 173 'Due to the faster scan time of TAFTS'->'Due to the faster scan time of ARIES' and to TAFTS where the scan time was corrected from 2s->1.5s.

Line 184: "...  the simulated FSI and FORUM-aircraft spectra produced similar results, suggesting a retrieval from an FORUM-aircraft observation is indicative of its FSI counterpart given a homogeneous scene".  I would recommend to add a few sentences explaining why the selected cloud is homogeneous and therefore the method is applicable to the studied case.

We are only saying that the results from this study will likely be representative of the FSI retrievals if the FSI sees a homogenous scene over it's footprint. The field of view of the aircraft-based instruments is significantly smaller than the FSI (TAFTS is approx. 250m and ARIES is approx. 400m) but persistent cirrus was measured by the lidar at the time of the radiance observations.

We have rewritten line 184 as follows to clarify:

"...the simulated FSI and FORUM-aircraft spectra produced similar results.  This suggests that the retrievals from the FORUM-aircraft observations investigated here will be indicative of what can be expected from the FSI if the latter is viewing homogeneous thin cirrus over its significantly larger footprint".

Figure 7: any comments on the pick of the uncertainty at around 750 cm$^{-1}$ and the BT residual for 35 um being consistently lower than for the rest of CRE?

Added to Figure 7 caption*: The peak in uncertainty at ~730 cm$^{-1}$ can be attributed to the FORUM-aircraft apodisation described in Section 2.3.2*

The BT residuals in Figure 7 come from a combination of the difference between the General Habit Mix and Solid Columns models and the impact of the Chou

approximation at each CER. To our knowledge, there are no explicit studies evaluating the BT difference between these two models for different ice clouds and atmospheres. While the BT residual for 35 microns is slightly lower than the other CER cases, the qualitative trend between the actual BT and CER for either habit model is as expected.

Line 325: "When the correct (GHM) bulk optical… ", what do you mean here by correct?

The GHM model is used to create the simulation and so is the correct model that the retrieval should infer. See the response to the Technical comment regarding Line 205 for further clarity about the FORUM-aircraft simulation.

Line 335: "This is an incorrect result as the GHM model was used in the simulation". I don't understand this sentence.

This result is inconsistent with the use of the GHM model in the simulation as described in Section 2.3.2. See response to Line 205 for additional clarification provided around the simulation.

Line 333-342: here it is discussed that the results using the GHM model are better, however, when looking at Table 3, MIR+FIR (SC) is the combination that delivers the closest results to the considered true state. I would suggest expand the discussion about GHM vs SC.

Added in:

*Table 3 shows that including the FIR leads to a closer agreement between all of the retrieved values and the known truth, as well as a significant reduction in the uncertainty associated with the CTH. The MIR+FIR GHM retrieval does show a slight positive bias in the cloud retrieval targets that is not seen with the MIR+FIR SC retrieval, as a result of reducing the spectral residual caused by the Chou approximation. However, it is important to recall that the retrieval is assessed using the measurement cost and spectral fit, and that the MIR+FIR GHM retrieval still captures the true state within 2 ESD.*

Line 357-360: I would add a few lines about why the observations exceed the uncertainty but not the simulation.

A clearer description has been included in the initial description of the simulations (see later response to comment for Line 205). The following has been edited:

*However, the BT residuals within the FIR generally exceed the measurement uncertainty, particularly between 450-500 $cm^{-1}$, while there is little change in the MIR residuals with and without the FIR. This was not seen in the retrieval from the FORUM-aircraft*

*simulation, which used the same bulk optical property model as the retrieval scheme. This reaffirms the findings in Bantges et al. (2020),...*

**Technical comments**

Line 79: this comment is a little bit picky, but what do you understand as a well characterized cirrus cloud?

A well characterized cirrus would have in-situ measurements of the cloud measuring the target properties (COT, CER, CTH etc.).

We do not state that this case contains a well characterized cirrus cloud, it is simply included as it is a target of the flight.

Figure 1: what is the difference between the continuous blue line and the dashed blue lines? If the first line was changed to continuous because of overlapping with the cloud extent, I would suggest marking the extent of the cloud with a continuous line, so all the in-situ are dashed and picking two colors that contrast a little bit more than blue and purple.

Line style corrected and colour changed

Line 33: add reference to some of the studies.

Added

Line 75: add reference to the website of CIRCCREX.

Added

Figure 2: I would add in the caption the altitude of the dashed blue lines so it is faster to identify which line corresponds to the the altitudes specified in Table 1.  "... have been characterised are shown in blue (6, 6.8, 7, 7.3, 7.5, 8., 8.3 km)"

Changed

Table 1: As a suggestion, I would add another column with the temperature.

Changed

Line 144: number distribution of sizes → size number distribution (?)

Changed

Line 147: I would recommend adding some references regarding the bimodality of cirrus clouds. For temperatures larger than approx. 210 K, bimodality starts playing a role. For lower temperatures, a fit to a monomodal PSD should be expected.

References added – we note that O'Shea et al, 2016 is specifically referred to here as this is the paper that describes the PSDs originally.

Line 179: for consistency, I would give the FOV of ARIES, TAFTS and FSI in the same units.

Changed to:

*We note that the ground footprint for the FSI (15 km) is greater than for TAFTS (~0.26 km) and ARIES (~0.41 km).*

Line 180: although the acronyms of NEDT and ARA appear in the caption of Fig. 3, I would suggest to write 'The apodised noise-equivalent-differential-temperature (NEDT) and target absolute radiometric accuracy (ARA)'.

Changed

Line 205: here is mentioned for the first time the "FORUM-aircraft simulation", but it passes a little bit unnoticed. For the rest of the manuscript it gets a bit confusing what the FORUM-aircraft observation is and what the FORUM-aircraft simulation is. I would suggest to rephrase this line to make clearer what the differences between the two are. Also in the caption of Fig. 4 is written "FORUM-aircraft LBLDIS simulation", is this the same as FORUM-aircraft simulation? Same for caption in Fig. 8 and caption in Table 3 and in line 310 where is written "LBLDIS GHM FORUM-aircraft simulation".

The introduction of the FORUM-aircraft simulation at the start of Section 2.3.2 has been expanded:

*This high-resolution spectrum is treated in two ways. First, it has the FSI apodisation directly applied to it, creating an FSI simulated spectrum. Second, the high-resolution spectrum is modified to look like TAFTS and ARIES before the FSI apodisation is applied to emulate the process applied to the observation. This creates a FORUM-aircraft simulated spectrum which will be used for later analysis in Section 4.1.*

For clarity it has been briefly reintroduced at the start of Section 4.1 (retrieval from the simulation):

*The FORUM-aircraft simulation described in 2.3.2 was simulated using the LBLDIS radiative transfer model, based on the profiles in Figure 2 and assuming the GHM model.*

The references to the simulation have all been changed to the FORUM-aircraft simulation for consistency

Line 216: ... and in the presence of cloud → ... and in the presence of clouds (?)

Changed

Line 245: "the ice cloud is modelled as a Gaussian of 1 km thickness". Is 1 km at full-width-halfmaximum?

Changed – 1km is the standard deviation of the Gaussian

Line 247, eq. 7: isn't a dz missing? And are the limits from 0 to z or from CBH to CTH? Since the ice crystal is much larger than the wavelength, is <Qe,vis> assumed to be 2?

Changed

Line 278: for convenience for the reader, I would add also here the five CER and the number for the fixed IWP.

Changed to:

*Figure 7 shows the brightness temperature (BT) residuals within these channels between the GHM and SC models for five different CER (25, 30, 35, 40, and 45 μm) and a fixed IWP of ~28 gm−2*

Figure 7: in the caption I would specify again of what the black line is uncertainty of.

Changed to:

*The black line represents the instrument and apodisation uncertainty and is the square-root-diagonal of the covariance matrix in Figure 5.*

Line 372: "…, before Section 5.3 evaluates the …" → "… and Section 5.3…"

Changed

Line 379: verticle → vertical

Changed

Line 399: for those not familiar with the term, I would specify "DOFS (degrees of freedom)".

This term is introduced at the end of Section 3.3.

Caption of Fig. 9, lines 2-3: I would specify Figure 8b. Also I would add ":" after "…as follows" and I would change the "." for "," in between SC(MIR): 0.33. GMH()…

Changed

Line 423: These have → It has (?)

Changed to 'The MODIS data have' for clarity

Line 431: do you mean from 0.9 to 1?

Changed

Caption of Fig. 11, in line 4: '... or CER. The error bars The measurements of the cloud taken...' → delete 'The error bars'. In line 5: ... from lidar measurements between from 09:48:09 to 09:49:09 → delete 'between' or delete 'from' and change 'to' for 'and'.

'The error bars' removed and 'from' removed

Line 454: "... 1600 cm-1" → $cm^{-1}$

Changed

The following comments related to references have also been changed

Line 484: https://doi.org/10.24381/cds.bd0915 doesn't work.

Line 487: "The MODIS data is taken from" this sentence is not finished.

Line 528: https://doi.org/https://doi.org/10.1016/j.jqsrt.2022 – delete the first https://doi.org/

Line 530: '...Cycle Coopera-tive' → '...Cycle Cooperative'

Line 548: https://doi.org/https://doi.org/10.1029/2022GL099394, e2022GL099394 2022GL099394, 2022 → https://doi.org/10.1029/2022GL099394, 2022

Line 564: I don't find the technical report.

Line 601: https://doi.org/https://doi.org/10.1029/2021JD035733, e2021JD035733 2021JD035733, 2022 → https://doi.org/10.1029/2021JD035733, 2022

Line 631: add doi: https://doi.org/10.1175/JAMC-D-11-067.1

Line 638-639: Delete " and add doi: https://doi.org/10.1175/JAS-D-12-039.1

**References**

Krämer, M., Rolf, C., Spelten, N., Afchine, A., Fahey, D., Jensen, E., Khaykin, S., Kuhn, T., Lawson,
P., Lykov, A., Pan, L. L., Riese, M., Rollins, A., Stroh, F., Thornberry, T., Wolf, V., Woods, S., Spichtinger, P., Quaas, J., and Sourdeval, O.: A microphysics guide to cirrus – Part 2: Climatologies of
clouds and humidity from observations, Atmos. Chem. Phys., 20, 12569–12608, https://doi.org/10.5194/acp-20-12569-2020, 2020.

Panditharatne, S., Brindley, H., Cox, C., Siddans, R., Murray, J., Warwick, L., and Fox, S.: Retrievals of water vapour and temperature exploiting the far-infrared: application to aircraft observations in preparation for the FORUM mission, Atmos. Meas. Tech., 18, 717–735, https://doi.org/10.5194/amt18-717-2025, 2025.

Saito, M., Yang, P., Huang, X., Brindley, H. E., Mlynczak, M. G., & Kahn, B. H. (2020). Spaceborne middle- and far-infrared observations improving nighttime ice cloud property retrievals. *Geophysical Research Letters*, 47, e2020GL087491. https://doi.org/10.1029/2020GL087491

**RC2 Comments**

**Exploiting airborne far-infrared measurements to optimise an ice cloud retrieval** by Panditharatne et al.

This is an interesting and valuable work. It describes the first retrieval of the cloud parameters, including optical depth, effective radius and top height, along with the atmospheric profiles of temperature and water vapour from upwelling far-infrared spectral radiances. In this work the RAL retrieval scheme, introduced in previous works, has been applied to the spectral radiance measurements performed with two spectrometers: the Tropospheric Airborne Fourier Tranform Spectrometer (TAFTS), operating in the far infrared (FIR) portion of the spectrum between 80-600 cm-1, and the Airborne Research Interferometer Evaluation System (ARIES) operating in the mid infrared (MIR) between 550-3000 cm-1. These measuremenets were performed on board of the aircraft B895 which flew in 2015 as part of CIRCCREX campaign. Carrying on board several instruments for cloud and atmosphere characterization, such as a series of three probes or CPIs and a backscattering/depolarization lidar to assess the cloud particle habit/size distributions and the extinction coefficient along with Vaisala probes providing information about humidity and temperature. The availability of such in situ measuremens allowed the comaparison with the retrieval products. The results show in generally a good accordance but also point out on the necessity of a refinenement of the currrent ice optical properties in models particularly in the FIR. Furthermore, it is shown the improvement in the retrieval performance by using the FIR portion with respect to neglect it, in particular in distinguishing the different crystal habits.

The manuscript is well written and structured even though some minor revisions need to be addressed before publication. I also suggest few corrections to enhance the clarity:

We thank the reviewer for their comments and have made the following changes to improve the clarity.

**Abstract:**

line 5  ":with and without the far infrared.." → "including and neglecting the far infrared portion of the spectrum.."

Changed

line 17:  "..cover.." → "..cover permanently.."

The reference I have used for this does not explicitly say permanently.

Line 54: "This work" → "The present work.."

Not changed

**Section 2.2**

Please, can you structure the initial part of the section by listing the in situ instruments present on board for cloud characterization, describing briefly what they provide? I think it is more simple for a reader to follow the text having  the scheme  in mind.

The following addition has been made to the start of Section 2.2:

*There were a number of instruments onboard the FAAM aircraft used to characterise the cirrus cloud that will be used to assess the retrievals performed here. These include an elastic backscatter lidar (used to derive a reference for the COT and CTH), a 2-DS probe (used to derive a reference for the CER), a 3-View Cloud Particle Imager (3V-CPI) (used to derive the ice crystal habits in O'Shea et al. (2016)), and a CIP 100 and a holographic cloud probe (HALOHolo) (used to measure the particle sizes).Further details about each of the instruments and their measurements are provided below.*

You say that the extinction coefficient profile obtained from lidar signal in Fig. 2a  is not reliable for deriving optical depth due to calibration issues; if you don't use it to calculate the COT, maybe, this is a little bit misleading,  wouldn't be better to show the raw signal?

In Section 2.3.2, the extinction coefficient profile is used to perform the LBLDIS simulation following the work in Bantges et al 2020 (where the extinction coefficient profile is scaled to match a best-fit COT that has been determined using a minimization process). In this way, it is useful to have the extinction profile in Figure 2. However, we agree it would be useful to show the range-corrected signal that is used for the COT

derivation, and so an additional subplot has been added to Figure 2 showing the range-corrected signal.

Regarding this, the COT calculated in Eq. (1) with the new method, is it obtained using the raw signal? I mean, are the Patt and Pref raw signals? Maybe it is better to report this in the text. If so, this is a reason to show the raw signal. in Fig. 2.

The range-corrected signal is used not the raw signal. The following is added:

*This approach considers the power of the attenuated Rayleigh backscatter range-corrected signals below 6 km*

However, how much the COT differ from that one calculated from the extinction profile? By eye, it does not look too far from the value you found, if we multiply the thickness for the mean extinction, but maybe I'm wrong. How the extinction coefficient has been derived from the backscattering signal?

The COT calculated from the extinction profile was 0.584. While not significantly different, this is a distinct underestimation that was shown in Bantges et al 2020 to limit our ability to simulate a spectrum within the TAFTS and ARIES uncertainty.

Vertical profiles of the particle extinction coefficient were determined using the Fernald-L=Klett method as described in Fox et al.,2019. The following statement has been added to the manuscript:

*…and a vertical profile of the particle extinction coefficient using the method described in Fox et al. (2019).*

The extinction due to the molecular contribution is already considered in the new method?

Yes, in this method the molecular extinction is considered but assumed to be constant along the short flight path, with signal variation attributed to particle (cirrus clouds) extinction. For clarity, the following sentence has been added:

*It is assumed that the molecular extinction is constant along the short flight path, and so the signal variation can be attributed to the ice crystals in the cirrus cloud.*

Caption Fig. 2: please indicate that are dashed lines.

Changed

Fig. 2 : the blue lines shouldn't be dashed?

Changed

I would suggest , if possible, to show some pictures of the habit and size distributions provided by the CPIs. It is just a suggestion because it would be very interesting to see

the in situ measuremenets of the ice crystals inside cirrus. And of course, I think, it would be an added value.

Images from the CPI and the size distributions have been provided in O'Shea et al (2016) and so are not provided here.

Section 2.2.1

Line 145: can you indicate the size range of the dominant smaller crystal?

Added: *Due to the dominance of smaller crystals (<30 µm),*

Can you show which parameters are fitted of the PSDs and report some results of the fit? Or briefly explain the procedure?

Information has been added about the procedure here:

*Following the findings of O'Shea et al. (2016), we find n(D) by fitting the PSDs measured by the Cloud 2-DS probe with bimodal Gaussian distributions using the non-linear least squares method. The fitted PSDs and parameters are shown in Figure S1 and Table S1 in the Supplement… The uncertainty presented here for the CER is calculated using the relative standard deviation of the fitting parameters, and so represents one standard deviation in the fitting of the bimodal Gaussian distribution to the PSD.*

The parameters fitted to the PSDs and results of the fit can be seen in the Figure S1 and Table S1 in the Supplementary information.

Line 148: "..the habit and.." → "..the habit type and.."

Changed

Line 148: "..CPI measurements habit recongnition.." →  "..CPI measurements by applying habit recongnition.."

Changed to: CPI measurements using the habit recognition algorithm

Can you report, if possible, a numerical example of the habit fractions derived from CPIs by applying the recognitional algorithm?

The exact numbers are shown in Figure 6b in O'Shea et al (2016) but the following has been added:

*Particles were found to be a mix of predominantly aggregates (∼30 %) and droxtals (∼25 %), with some rosettes (∼15 %) and columns (∼10 %).*

**Section 2.3.1**

Line 186:  with "remove" you mean deconvolved?

Changed

Line 191 and 193: Just remember that are available more recent version of LBLRTM and, in particular, of continuum MT_CKDv3.8

Thank you for your comment. We note that we use MT_CKDv3.5 which has a significant update to the water vapour spectroscopy in the far-infrared, but of course is still an older version.

Line 199: "..using the Masuda model above 769cm-1" → "..using the Masuda model above 769cm-1 since the radiometric measurements were performed over the Ocean" (Is it correct? If so I would clarify why the Masuda model has been used)

*'as the radiance observation was taken over the ocean'* has been added

Line 203: "and the GHM model" → "assuming the GHM model"

Changed

Fig. 4: The differences of TAFTS in blue are not visible, would be possible to expand the scale just for this plot of TAFTS for example in logarithmic scale to make it more readable?

The difference with TAFTS is significantly smaller than it's ARIES counterpart, and when a log scale is used (or a logscale is used only for TAFTS), the plot implies that the two are both noteworthy while only the differences for ARIES are significant. To avoid misinterpretation, a version of Figure 4 has been included in the supplementary information (Figure S2) with a smaller scale used for the TAFTS residuals shown in Figure 4a, and the following has been added to the caption in Figure 4:

*(a) The brightness temperature (BT) residual between the LBLDIS simulation of the TAFTS and ARIES observations during the B895 flight that has had the FORUM apodisation directly applied and been made to look like TAFTS or ARIES first (FORUM-aircraft simulation) as outlined in Section 2.3.2. The TAFTS residuals remain less than 0.05 K and can be seen in an additional plot in the Supplement for clarity.*

**Section 3**

Could you indicate what is the initial guesses or if they are equal to a-priori?

Added: *'In all the retrievals performed in this work, the initial guess is equivalent to the a-priori'*

Which are the correlation lengths used for a-priori atmospheric profiles?

The correlation lengths are not explicitly defined in this retrieval scheme. Instead, the a-priori covariance matrix is used to define correlations between atmospheric layers with its off-diagonals.  This is described in the Siddans, et al 2019. which is referenced when the retrieval scheme is introduced.

Eq. (7) the differential dz is missing inside the integral.

Changed

If you fit the CTH how much did you fix the geometrical thickness?

Clarified: Within IMS, the ice cloud is modelled as a Gaussian with a standard deviation of 1 km.

Line 431:   maybe you mean "..increase from 0.9 to 1.0"? not from 0.9 to 0.1

Changed